# The Nuclear Transporter Importin 13 Can Regulate Stress-Induced Cell Death through the Clusterin/KU70 Axis

**DOI:** 10.3390/cells12020279

**Published:** 2023-01-11

**Authors:** Katarzyna A. Gajewska, David A. Jans, Kylie M. Wagstaff

**Affiliations:** Biomedicine Discovery Institute, Monash University, Clayton 3168, Australia

**Keywords:** nuclear transport, cellular stress, cell death, importin 13

## Abstract

The cellular response to environmental stresses, such as heat and oxidative stress, is dependent on extensive trafficking of stress-signalling molecules between the cytoplasm and nucleus, which potentiates stress-activated signalling pathways, eventually resulting in cell repair or death. Although Ran-dependent nucleocytoplasmic transport mediated by members of the importin (IPO) super family of nuclear transporters is believed to be responsible for nearly all macromolecular transit between nucleus and cytoplasm, it is paradoxically known to be significantly impaired under conditions of stress. Importin 13 (IPO13) is a unique bidirectional transporter that binds to and releases cargo in a Ran-dependent manner, but in some cases, cargo release from IPO13 is affected by loading of another cargo. To investigate IPO13′s role in stress-activated pathways, we performed cell-based screens to identify a multitude of binding partners of IPO13 from human brain, lung, and testes. Analysis of the IPO13 interactome intriguingly indicated more than half of the candidate binding partners to be annotated for roles in stress responses; these included the pro-apoptotic protein nuclear clusterin (nCLU), as well as the nCLU-interacting DNA repair protein KU70. Here, we show, for the first time, that unlike other IPOs which are mislocalised and non-functional, IPO13 continues to translocate between the nucleus and cytoplasm under stress, retaining the capacity to import certain cargoes, such as nCLU, but not export others, such as KU70, as shown by analysis using fluorescence recovery after photobleaching. Importantly, depletion of IPO13 reduces stress-induced import of nCLU and protects against stress-induced cell death, with concomitant protection from DNA damage during stress. Overexpression/FACS experiments demonstrate that nCLU is dependent on IPO13 to trigger stress-induced cell death via apoptosis. Taken together, these results implicate IPO13 as a novel functional nuclear transporter in cellular stress, with a key role thereby in cell fate decision.

## 1. Introduction

Regulated transport of proteins between the nuclear and cytoplasmic compartments of the eukaryotic cell through the nuclear pore is fundamental for normal cellular function, tightly regulating critical processes, including gene expression, signal transduction, stress responses, cellular proliferation, differentiation, and development. Proteins larger than ~40 kDa require an active process, conventionally facilitated by members of the importin (IPO) superfamily of transport proteins, of which there are multiple α and β types [1,2,3,4]. IPOβ family members consist of IPOs that import proteins into the nucleus and exportins (XPOs) that export proteins out of the nucleus. Conventional nuclear transport mediated by the IPOβ family is critically dependent on the small GTP-binding protein Ran. The dissociation of an import complex or the formation of an export complex within the nucleus are both dependent on RanGTP binding to IPOβ, while dissociation of an export complex in the cytoplasm relies on hydrolysis of RanGTP to RanGDP [1,2,5,6]. This cycle establishes a Ran concentration gradient across the nuclear envelope that is crucial to all conventional nuclear transport, whereby RanGTP is highly concentrated within the nucleus but is low in the cytoplasm.

Environmental stresses that disrupt cell homeostasis, including those induced by oxidative stress and heat shock, trigger cellular stress responses that are largely potentiated by transcription factors which traffic between the cytoplasm and the nucleus in order to regulate the appropriate transcriptional response [7,8]. Somewhat paradoxically, these same stresses have also been established to induce changes to the localisation of key nuclear transport machinery components, resulting in significant inhibition of conventional IPO-dependent nuclear trafficking [7,9,10]. Central to this phenomenon is the stress-induced collapse of the Ran gradient, where nuclear RanGTP levels are depleted, and Ran is almost entirely mislocalised to the cytoplasm [10,11]. The loss of nuclear RanGTP is thought to trigger the subsequent mislocalisation of IPO proteins, including IPOβ and IMPα proteins [10,12,13,14,15], with conventional nuclear import and export consequently affected [16,17,18,19]. Given that cellular stress responses require active nuclear transport of transcription factors but conventional Ran-dependent pathways are not functional under these conditions, alternative pathways must exist to mediate these critical cell responses to stress.

IPO13 is a unique member of the IPOβ family and is one of only three mammalian IPO proteins able to mediate both import and export of cargo proteins [20,21,22,23]. Nuclear import cargoes of IPO13 include the E2 SUMO-conjugating enzyme UBC9 [24], paired-type homeo domain transcription factors Pax6, Pax3, and Crx [25], Aristaless-related homeobox gene (ARX) [26], and the glucocorticoid receptor (GR) [27], while nuclear export cargoes include eukaryotic translation initiation factors 1A (EIF1A) [21] and 4G2 (EIF4G2), the high mobility group containing protein 20A (HMG20A) [28] and transcription factors Kruppel-like factor 4 (KLF4) and specificity protein 1(SP1) [29]. By facilitating nucleocytoplasmic protein transport of these and other cargo, IPO13 contributes to key developmental processes, such eye morphogenesis, and testis [30], brain [31,32], and lung development [33,34]. More recently, we established that IPO13 plays a role in the transcriptional response to oxidative stress [29,35]. Interestingly, the requirement of IPO13 for RanGTP to bind/release certain cargo is not absolute. In some cases, hydrolysis of RanGTP to RanGDP is not sufficient to mediate export cargo release from IPO13, but instead, an import cargo is required to competitively displace any remaining bound export cargo and subsequently bind to IPO13 prior to import [21,24].

To investigate the biological role of IPO13 in various key tissues, we performed a yeast 2-hybrid (Y2H) screen using libraries from human brain, lung, and testis, identifying 62 potential binding partners. Strikingly, these included a high proportion of proteins involved in the cellular stress response, including the pro-apoptotic protein nuclear clusterin (nCLU) and the DNA repair protein KU70. nCLU is known to induce apoptosis in response to stress, in part through enhanced nuclear accumulation and interaction with KU70 under these conditions, which appears to reduce KU70 DNA binding [36,37,38]. We confirmed binding between IPO13 and both of these proteins in a cellular context and determined that IPO13 mediates their nuclear import and export respectively. Significantly, by promoting nuclear localisation of nCLU in response to stress and, in turn, reducing the export of KU70, IPO13 contributes to apoptotic cell death induced by nCLU, in part, through DNA damage/interference with DNA repair. Finally, we confirm that, unlike several other IPOs, IPO13 continues to traffic efficiently under conditions of stress, establishing a unique nuclear transport pathway that is active under conditions of stress, the first such pathway mediated by an IPO to be identified.

## 2. Materials and Methods

### 2.1. Yeast-Two-Hybrid Screen and Functional Annotation Analyses

Yeast-2-hybrid (Y2H) screening was performed in collaboration with Hybrigenics Inc. (Paris, France), using sequences encoding amino acids 1-963 of IPO13 as bait and cDNA libraries derived from human lung, brain, and testis. The coding sequences of positive clones were amplified by PCR and sequence analysis performed to identify the protein. Positive clones were ranked, as previously [39], on the basis of an algorithm-based confidence score (Predicted Biological Score, PBS), where the score represents the probability of the interaction being nonspecific, ranging from A, which represents the highest confidence in the interaction, to D, representing the lowest confidence in the interaction. A score of E denotes proteins that were identified in more than 10 Y2H screens and is indicative of proteins able to interact with a wide range of proteins. Pathways and functional GO term enrichment analysis was performed using DAVID Bioinformatics Resources 6.8, using the Functional Annotation Tool, where proteins, irrespective of any other biological functions, were sorted into major functional classes on the basis of co-occurrence with the rest of the proteins identified in the Y2H.

### 2.2. Plasmid Constructs

Plasmids containing the ORF of KU70 and CLU in pentr322 vectors were a gift from Dr. Travis Johnson (Department of Biochemistry and Molecular Biology and School of Biological Sciences, Monash University, Melbourne, Australia). The pEGFP-KU70 expression vector was constructed by inserting the KU70 coding sequence PCR amplified from pentr322-KU70 into the Sal1 and BamH1 restriction sites of pEGFP-C1 Vector (Clontech, Mountain View, CA, USA), including a TAA stop codon directly after KU70. The pEGFP-nCLU and mCherry-nCLU expression vector were constructed by amplifying nCLU from the second ATG initiation sequence within the ORF of pentr322-CLU and inserting it into the HindIII and BamH1 restriction sites of pEGFP-C1 and mCherry-C1. The plasmid GFP-IPO13 was used for mammalian cell expression of GFP-IPO13, as described previously [28], and the plasmid pEGFP-C1-Importin7 for mammalian cell expression of GFP-IPO7 was donated by Eldar Zehorai (Weizmann Institute, Rehovot, Israel). The pDsRed2-C1-LIPO13 expression vector was constructed by inserting PCR-amplified IPO13 into the EcoRI and SmaI restriction sites of vector pDsRed2-C1 (Clontech, Mountain View, CA, USA). The pIRES-mCherry-LIPO13 expression vector was constructed by inserting PCR-amplified IPO13 into the Not1 restriction sites of the vector pCMV-IRES-mCherry, described previously [28]. pEPI-XPO1 was generated using the Gateway^®^ BP Clonase System (Invitrogen, Waltham, MA, USA). pEGFP-IPOβ1 and pEGFP-IMPα2 were a gift from Yoichi Miyamoto. Bacterial expression plasmids encoding His_6_-GFP-KU70 or His_6_-GFP-nCLU were generated via the Gateway^TM^ Cloning System (Invitrogen, Waltham, MA, USA). Briefly, KU70 was PCR-amplified from pentr322-KU70 and nCLU from pentr-322CLU. These were recombined (via the “BP” reaction) into a pDONR207 vector (Invitrogen, Waltham, MA, USA), and the resultant pDONR207-KU70 and pDONR207-nCLU vectors were recombined (via the “LR” recombination reaction) into pGFP-attC [40]. Bacterial expression plasmid pGEX4T2-IPO13 was generated as described [41].

### 2.3. Cell Culture and Transfection/Immunofluorescence

HeLa and OHIO HeLa human cervical cancer cells were cultured in Eagle’s Minimum Essential Medium (EMEM) containing 10% FBS, 1x Glutamax, and 1x Nonessential Amino Acids (NEAA) at 37 °C in 5% CO_2_ atmosphere in a humidified incubator. IPO13^+/+^ and IPO13^−/−^ mESC lines, previously described in detail [28], were maintained in feeder-free culture conditions in Dulbecco’s Modified Eagle’s Medium (DMEM) supplemented with 12.5% FBS, 1x Glutamax, 0.1 mM NEAA, 0.1 mM β mercaptoethanol, and 1000 U/mL Leukemia inhibitory factor (LIF) at 37 °C on 0.1% gelatin-coated surfaces in 5% CO_2_ atmosphere in a humidified incubator. mESC cells were passaged every 2 d. Plasmid transfection was performed using FuGene^®^ HD Transfection Reagent (Promega, Madison, WI, USA; HeLa cells) or Lipofectamine^®^ 2000 (Invitrogen, Waltham, MA, USA; IPO13^+/+^ and IPO13^−/−^ mESCs) according to manufacturer’s instructions. The Dharmacon ON-TARGETplus siRNA system (GE Healthcare, Lafayette, CO, USA) with DharmaFECT 1 transfection reagent was utilised as per manufacturer’s instructions using pre-designed siRNA-targeting IPO13 (SMARTpool L-020212-01) and a non-targeting (NT siRNA) control pool (D001810-10-50) as the negative control. For immunofluorescence microscopy, cells were immunostained as previously described [42]. Briefly, cells grown on glass coverslips were fixed using 4% paraformaldehyde (PFA) and permeabilised with 0.2% Triton-X-100 (Sigma, St. Louis, MO, USA). Non-specific binding sites were blocked with blocking buffer (1% BSA in phosphate-buffered saline (PBS)), and cells were immunostained with primary antibodies at room temperature for 1 h followed by 1 h incubation with conjugated secondary antibodies. Coverslips were mounted to glass slides using ProLong Gold antifade reagent with DAPI (Invitrogen, Waltham, MA, USA). Antibodies used for immunostaining were mouse monoclonal anti-Ran (Cat no. 610340, BD Biosciences, Franklin Lakes, NJ, USA), mouse anti-UBC9 (Cat No. 610759, BD Biosciences, Franklin Lakes, NJ, USA) rabbit polyclonal anti-EIF1A (ab38976, Abcam, Cambridge, UK), and Alexa-Fluor 488-donkey anti-mouse, Alexa-Fluor 568-goat anti-mouse, and Alexa-Fluor 568-goat anti-rabbit IgG secondary antibodies (Invitrogen, Waltham, MA, USA).

### 2.4. Protein Expression and Purification

IPO13 was purified from E.coli Bl21 bacteria as a GST-fusion protein using glutathione (GSH) agarose beads (GE Healthcare, Lafayette, CO, USA) [41]. GST-IPO13 was biotinylated and concentrated as previously described [43]. His-tagged GFP-fusion proteins were expressed under similar conditions to those described previously [44]. *E. coli* (BL21 pRep4 strain) was transformed with constructs encoding p-attC-KU70 or –nCLU and grown at 37 °C until mid-log phase, followed by induction using 1 mM isopropyl-1-thio-β-D-galactopyranoside (IPTG) for 5 h at 37 °C. Proteins were purified using nickel–nitrilotriacetic acid beads (Qiagen, Hilden, Germany) under denaturing conditions (8 M urea) and then renatured on the column and eluted with 200 mM imidazole before dialysis and concentration using a 50 KDa MWCO spin column (Amicon, Millipore, Burlington, MA, USA). Proteins were further purified by fast protein liquid chromatography (FPLC) using size exclusion columns.

### 2.5. AlphaScreen Importin-Binding Assay

An AlphaScreen-based binding assay was used to determine the binding affinity of His-tagged proteins to biotinylated IPO13, as described previously [43]. His_6_-GFP fused proteins (at 30 nM final concentration) were incubated with 0–60 nM biotinylated GST or GST-IPO13. AlphaScreen counts were measured on an EnSpire (PerkinElmer, Waltham, MA, USA) plate reader. Triplicate values were averaged, and sigmoidal association curves were plotted using Prism 7.

### 2.6. Confocal Laser Scanning Microscopy (CLSM) and Image Analysis

Cells immunostained for endogenous protein or expressing fluorescently tagged protein were imaged at 60× magnification using a FluoView^TM^ FV1000 Confocal Microscope. Imaging in live cells was performed in an FCS2 live-cell chamber maintained at 37 °C (Bioptechs, Butler, PA, USA). The nuclear-to-cytoplasmic-fluorescence ratio (Fn/c) was calculated according to the equation Fn/c = (Fn − Fb)/(Fc − Fb), where Fn is the nuclear fluorescence, Fc is the cytoplasmic fluorescence, and Fb is the background auto fluorescence from digitised images using ImageJ software (NIH). The nuclear envelope (NE)-to-nuclear-fluorescence ratio (Fne/n) was calculated in a similar fashion according to the equation Fne/n = (Fne − Fb)/(Fn − Fb), where Fne is the nuclear envelope fluorescence. The Fne was measured using the straight-line tool at a line width of 5 using the ImageJ software (NIH).

### 2.7. Fluorescence Recovery after Photobleaching (FRAP) Approach to Analyse Nuclear Transport

HeLa cells transfected to express GFP-tagged proteins were imaged live using an Olympus Fluoview1000 CLSM with a 60× oil immersion objective at 37 °C. As previously described [45], regions of interest were photobleached (12 scans, 12.5 s/pixel, with high laser power), and the fluorescence recovery was observed (low laser power, 10 s/pixel) at 20 s intervals for 460 s. The relative level of fluorescence in the bleached and unbleached regions was quantified using ImageJ, measuring the return or loss of nuclear fluorescence resulting from import or export respectively of fluorescently tagged protein into or out of the nucleus. Data are presented as fractional recovery (Frec) of nuclear fluorescence (minus background) relative to the pre-bleached values, calculated according to the formula: Frec(Fn − b) = (Fn − Fb)/prebleach fluorescence, where Fn is nuclear fluorescence and Fb is background fluorescence. Relative level of nuclear accumulation/loss at each time point was quantified to determine the fractional gain or loss of nuclear fluorescence Frec(Fn − b), the initial rate was determined as a change in the Frec (Fn − b)/s for the liner portion of the curve, and the half time for maximal (t_1/2_) fluorescent loss/recovery was quantified by plotting individual values for Frec(Fn − b) (y) against time (t) using a one-phase association nonlinear equation in Prism 7.

### 2.8. Co-Immunoprecipitation and Western Analysis

HeLa cells co-expressing GFP or GFP-nCLU proteins with DsRed2-IPO13 were treated with or without 125 μM H_2_O_2_ (Sigma, St. Louis, MO, USA) for 1 h prior to cell lysis. HeLa cells expressing GFP or GFP-IPO13 protein treated with or without 125 μM H_2_O_2_ (Sigma, St. Louis, MO, USA) for 1 h underwent protein cross-link using Gluteraldehyde (Sigma, St. Louis, MO, USA) as previously described [46], after which they were lysed immediately. In both cases, immunocomplexes were precipitated and eluted using GFP-Trap resin (ChromoTek, Proteintech, Rosemont, IL, USA) according to the manufacturer’s instructions. Input and immunoprecipitation fractions were analysed by Western analysis using anti-IPO13 (Cat no. 11696-2-AP, Protein Tech, Rosemont, IL, USA), anti-KU70 (Cat no. ab3114, Abcam, Cambridge, UK), anti-GFP (Cat no. 11814460001, Roche, Basel, Switzerland), anti-Actin (Cat no. ab3280 Abcam, Cambridge, UK), and β-actin (CST, Cat no. 3700) primary antibodies and the secondary antibodies goat anti-mouse IgG HRP (AP308P) and goat anti-rabbit IgG HRP (AP307P).

### 2.9. In-Gel Neutral Comet Assay

In-gel neutral comet assays were performed essentially as described [47,48]. HeLa cells (with either siRNA treatment or transfected to express GFP-tagged proteins) were trypsonised and resuspended in PBS at 1 × 10^5^ cells/mL, then diluted 1:10 with 1% low-melt-point agarose. The amount of 40 μL cell suspension was pipetted per well onto pre-coated 20-sample CometSlide^TM^HT slides (Trevigen, Gaithersburg, MD, USA) and incubated at 4 °C for 30 min, followed by 50 μM H_2_O_2_ for 1 h at 4 °C in culture medium, where required for stress treatment. After stress treatment, slides were washed with cold PBS and one slide placed in Comet Lysis Solution (2.5 M NaCl, pH 10, 100 mM EDTA, 10 mM Tris, 1% Sarkosyl, 1% Triton X) for 1 h at 4 °C. Where appropriate, a duplicate slide was placed in fresh cell culture medium and incubated at 37 °C for a 2 h recovery before lysis as above. All slides were then incubated in 1 x neutral electrophoresis buffer (TBE) for 30 min prior to electrophoresis in fresh buffer at 1 V per cm (between each electrode) for 40 min at 4 °C in the dark. Slides were washed in pre-chilled dH_2_O for 5 min, followed by 5 min in pre-chilled 70% EtOH and dried overnight away from direct light. Slides were then stained with 1 x Sybr Green (Sigma, St. Louis, MO, USA) for 20 min, rinsed in dH_2_O, and dried again. Comets were imaged on the Olympus Provis AX70 Widefield system and analysed using the ImageJ analysis plugin OpenComent 1.3 to determine the percentage of DNA in the comet tail of >100 comets per sample.

### 2.10. Flow Cytometry

IPO13^+/+^ and IPO13^−/−^ ESCs were transfected with GFP-nCLU or GFP alone 24 h prior to treatment with 125–600 μM H_2_O_2_ for 1 h as appropriate. Cells were harvested by trypsonisation/centrifugation, resuspended in 0.5 mL ice-cold PBS supplemented with 12.5% FBS, and stained with PI (Invitrogen, Waltham, MA, USA) at a final concentration of 1 μg/mL. PI-positive cells within the GFP-transfected or GFP-nCLU cell populations were quantified by flow cytometry (FACS Calibur, BD Biosciences, Franklin Lakes, NJ, USA), with a minimum of 15,000 cells analysed per sample. Data analysis was performed using the FlowJo software. Statistical analyses were carried out using two-way ANOVA. Annexin V/PI double-staining was performed to detect apoptosis in IPO13^+/+^ and IPO13^−/−^ ESC populations. Cells were treated with 12 μM Camptothecin (CTH) for 6 h prior to staining with FITC-labeled Annexin V and PI using a FITC-Annexin V apoptosis detection kit (BD Bioscience, Franklin Lakes, NJ, USA) according to the manufacturer’s instructions. A minimum of 15,000 cells were analysed with a FACS Calibur flow cytometry system and further analysed using FlowJo software.

### 2.11. Statistical Analysis

Statistical analysis was performed used Student’s *t*-test unless otherwise stated using Prism 7 software.

## 3. Results

### 3.1. The IPO13 Interactome

To shed light on IPO13 as a bidirectional nuclear transporter with apparent reduced dependence on Ran compared with other IPOβ family members [21,24], we performed a Y2H screen to identify novel binding partners. The Y2H screen was conducted using cDNA libraries from human brain, lung, and testis with full length IPO13 as bait (amino acids 1-963), resulting in the identification of 62 potential binding partners in total (Table 1). A number of these were identified in more than one of the libraries used. To the best of our knowledge, this represents the first Y2H screen performed with the full length IPO13 protein. Only two previous Y2H screens have been performed utilising IPO13 without its Ran-binding domain [25] or utilising the N-terminally truncated isoform of IPO13 [28]. Several of the proteins identified in our screen have previously been identified as IPO13 binding partners using diverse approaches (see Table 1), giving confidence to the rigour of our screen—RANBP1 and SF3B2, for example, have been identified by Mackmull et al. (2017) and Baade et al. (2018) using proximity ligation coupled to mass spectrometry (BioID) and SILAC and mass spectrometry, respectively. Importantly, our screen was also able to identify an abundance of novel IPO13 binding candidates, significantly expanding the IPO13 interactome.

Candidate binding partners were functionally annotated using the Functional Annotation Tool of the DAVID Bioinformatics Resources v6.8 (see Materials and Methods) [49,50], revealing an enrichment of proteins involved in the regulation of gene expression and proteins involved in nuclear transport with GO terms, including nuclear export, nuclear import, nuclear pore, and nuclear localisation sequence binding. Intriguingly, pathways analysis revealed an enrichment of proteins involved in the cellular stress response, with 58% of proteins annotated to GO terms, including response to stress, cellular response to DNA damage stimulus, and regulation of cellular response to heat (Table 1). It is also worth noting, given IPO13′s acknowledged role in brain development, that several binding partners were also annotated to neurogenesis. The interactome included validated IPO13 transport cargoes, such as Pax6, identified from the brain library, as expected, and HMG20A [25,28]. Both of these were scored at confidence category E in terms of the Predicted Biological Score (PBS), giving confidence overall to the total list of putative binding partners with comparable or higher PBS scores (see Table 1).

### 3.2. IPO13 Acts as a Nuclear Import Factor for nCLU and an Export Factor for KU70

That IPO13 appears to bind several proteins involved in the stress response implies that IPO13 could be responsible for transport of proteins involved in the cellular stress response, consistent with our recent findings that IPO13 regulates nuclear export of SP1 and KLF4 under conditions of oxidative stress [29]. To examine this prospect further, we screened 6 interactors for stress-induced changes in localisation; 3 known IPO13 cargoes (UBC9, ARX, and EIF1A) and 3 of the 36 stress-related cargo proteins from our Y2H screens (Pax6, CLU, and KU70) (Table 1). HeLa cells, with or without transfection to express fluorescently labelled protein, were treated with 125 μM hydrogen peroxide (H_2_O_2_) to induce oxidative stress, as previously [29,51,52]. Cells were imaged live or after being fixed and immunostained, with digitised images of the cells being analysed to determine the nuclear-to-cytoplasmic-fluorescence ratio (Fn/c, see Materials and Methods) of each cargo before and after stress, where an Fn/c value below 1 indicates nuclear exclusion and above 1 indicates nuclear accumulation. The nuclear localisation of the known IPO13 import cargoes UBC9 (Fn/c of 1.1 compared with 1.6) and GFP-ARX (Fn/c = 17 compared with 41) and export cargo EIF1A (Fn/c of 4 compared with 8) were all significantly (*p* < 0.0001) reduced by oxidative stress (Appendix A–f). Of the three Y2H hits, Pax6, confirmed here, is a known IPO13 import cargo, with treatment of oxidative stress also reducing nuclear accumulation of mCherry-Pax6 significantly (*p* < 0.0001) (Appendix A, Fn/c of approx. 19 compared with 40). Thus, all of the IPO13 cargoes tested were more cytoplasmic in the presence of oxidative stress, suggesting either inhibition of their nuclear import or enhanced nuclear export under stress conditions.

Next, we examined the localisation of DNA repair protein KU70 and Clusterin (CLU), both of which are novel cargo candidates with PBS scores of E and B, respectively, covering the range of confidence scores across the Y2H screen. KU70 and CLU are known to interact with each other and are both regulators of the same stress-induced intrinsic apoptosis/DNA repair pathway [37,38,53,54]. KU70 is a key component of the non-homologous end-joining (NHEJ) DNA repair pathway [55], while CLU plays a complex regulatory role in the cellular stress response, with differential transcription of the CLU gene producing two key isoforms with opposing roles: the anti-apoptotic secretory CLU (sCLU) [56] and the pro-apoptotic nuclear CLU (nCLU) [36,54]. In addition to its DNA repair roles in the nucleus, KU70 is known to interact with sCLU and the critical pro-apoptotic protein Bax in the cytoplasm to prevent Bax-dependent apoptotic cell death [53,57]. In response to cellular stress, the transcription of nCLU is upregulated; while it localises in the cytoplasm of untreated cells, under stress, nCLU accumulates within the nucleus, although the import receptor responsible had not yet been identified prior to the present study [37,58].

We determined the localisation of GFP-tagged nCLU isoform of CLU (both the nCLU and sCLU isoforms share the selective interactive domain mapped in Table 1) or KU70 in HeLa cells transfected to co-express DsRed2 or DsRed2-IPO13 in the absence or presence of oxidative stress treatment (Figure 1a–d). Consistent with previous reports [36], GFP-nCLU was predominantly cytoplasmic in most cells in the absence of H_2_O_2_. (Figure 1a), with only a small population of cells displaying some nuclear accumulation of GFP-nCLU. Digitised images were analysed, confirming strongly cytoplasmic localisation (Fn/c of 0.3). Oxidative stress significantly (*p* < 0.0001) increased nuclear accumulation of GFP-nCLU (Fn/c of 0.6). GFP-KU70, which was already highly nuclear (Fn/c of 21) in untreated cells, accumulated in the nucleus to a significantly (*p* < 0.0001) greater extent (Fn/c = 35) under stress conditions (Figure 1c,d). In contrast to the other cargoes tested, which showed reduced nuclear accumulation in response to oxidative stress, nCLU and KU70 were distinct in becoming more nuclear. To test whether this nuclear accumulation could be attributed to stress-enhanced IPO13-mediated nuclear import, we co-expressed DsRed2-IPO13 with GFP-nCLU, finding significant (*p* < 0.0001) nuclear accumulation in untreated cells (Fn/c = 0.3 vs. 0.5), suggesting that IPO13 can promote nuclear import of GFP-nCLU (Figure 1a,b). Interestingly, this increase was to a level similar to that observed in stress-treated cells in the absence of IPO13 over-expression; however, stress treatment did not further enhance the nuclear accumulation in the presence of IPO13 overexpression.

IPO13 overexpression is useful to examine IPO13 trafficking, as the concentration of IPO13 is rate-limiting for nuclear import and export (Baade et al., 2018), so overexpression can cause dramatic changes in cargo localisation. In contrast, overexpression of IPO7 had no effect on nCLU localisation (Appendix A). In contrast, ectopic expression of IPO13 in untreated cells significantly (*p* < 0.0001) reduced nuclear localisation of GFP-KU70, (Fn/c of 6), suggesting that IPO13 may act as a nuclear export protein for KU70 under normal conditions (Figure 1c,d). Interestingly, when cells overexpressing Ds-Red2-IPO13 were treated with H_2_O_2_, GFP-KU70 showed significantly (*p* < 0.0002) increased nuclear localisation (Fn/c of 11.5), but to a much lower level (*p* < 0.0001) than in cells not expressing DsRed2-IPO13 (Fn/c of 35). Since nCLU has been shown to bind to KU70 in the nucleus under conditions of stress [37], this result may reflect increased nuclear retention of KU70 due to increased levels of nuclear nCLU (see below).

To confirm these observations, we examined the effect of siRNA-induced knockdown of IPO13 on the localisation of GFP-nCLU and -KU70 in the presence and absence of H_2_O_2_ treatment. GFP-nCLU nuclear accumulation was significantly (*p* < 0.0001) increased in the presence of stress (Fn/c = 0.2 increased to 0.5) in cells treated with non-targeting (NT) siRNA. GFP-nCLU localisation was not significantly altered in response to stress in cells treated with IPO13-siRNA (Fn/c of approx. 0.3; Figure 1f,g), confirming that GFP-nCLU nuclear entry is, in part, dependent on IPO13 in the presence of stress. IPO13 knockdown in unstressed cells significantly (*p* = 0.0002) enhanced GFP-KU70 nuclear accumulation (Fn/c = 27 increased to 38); this level of accumulation was comparable to that observed in NT-treated cells under stressed conditions (Fn/c of 40), consistent with the idea that IPO13 mediates GFP-KU70 nuclear export under normal conditions (Figure 1h,i). Interestingly, knockdown of IPO13 significantly (*p* = 0.0031) reduced stress-induced nuclear accumulation of GFP-KU70 (Fn/c of 29) to levels close to basal in the absence of stress (Figure 1h,i); since IPO13 knockdown also reduces nCLU nuclear accumulation (Figure 1a,b,f,g), the reduced level of nuclear nCLU appears to result in reduced KU70 nuclear retention (see below).

To pursue this further, we tested for direct interaction of IPO13 with nCLU or KU70 and whether this interaction can occur in stressed cells. GFP-nCLU was expressed in HeLa cells in the absence or presence of H_2_O_2_ treatment (Figure 1j) with DsRed2-IPO13 and cell lysates subjected to immunoprecipitation (IP) using GFP-trap. Significantly, DsRed2-IPO13 co-precipitated with GFP-nCLU under both treated and untreated conditions, confirming that nCLU is complexed with IPO13 in a cell context even during cell stress. To assess interaction with KU70 protein, HeLa cells co-transfected to express GFP-IPO13 and mCherry-KU70 were treated with glutaraldehyde to crosslink protein complexes prior to cell lysis and IP using GFP-trap beads (Figure 1k). Both endogenous KU70 and mCherry-KU70 co-precipitated with GFP-IPO13 under both control and stress conditions, but to lower levels in H_2_O_2_-treated cells, consistent with the idea that KU70 interacts with IPO13 in a cell context but that this interaction is reduced by cellular stress. To confirm direct binding between IPO13 and nCLU or KU70, AlphaScreen binding assays with bacterially expressed proteins were used to determine the affinity of the interaction (K_D_). Both His_6_-GFP-tagged nCLU and -KU70 bound to GST-IPO13 with high affinity (K_D_ = 2.8 and 7.3 nM respectively, Figure 1l,m), confirming the ability of IPO13 to bind directly to both nCLU and KU70.

Under conditions of stress, nCLU and KU70 undergo complexation in the nucleus. This interaction may modify the binding activity of KU70 to DNA, as evidenced by KU70′s reduced DNA end-binding activity when CLU is overexpressed in cell lysate [37,38]. We speculated whether the interaction between nCLU and KU70 in the nucleus of stressed cells could also reduce IPO13-dependent nuclear export of KU70 under these conditions. To explore this, we tested the effect of nCLU overexpression on the KU70-IPO13 interaction; GFP-KU70 was expressed in HeLa cells in the absence or presence of oxidative stress and co-transfected with mCherry or mCherry-nCLU. After crosslinking using glutaraldehyde, GFP-KU70 was subjected to IP using GFP-trap beads (Figure 1n,o). Densitometric analysis indicated that the amount of IPO13 that was co-precipitated with GFP-KU70 under conditions of oxidative stress was significantly (*p* < 0.0006) reduced in cells co-transfected with mCherry-nCLU, compared with cells co-expressing mCherry alone, whereas there was no significant difference in the amount of IPO13 co-precipitated with GFP-KU70 in untreated cells co-expressing either mCherry-nCLU or mCherry alone. These findings suggest that nCLU reduces the KU70-IPO13 interaction under conditions of stress. To begin to examine whether this is the result of nCLU sequestering KU70 from IPO13 under these conditions, we tested the effect of GFP-KU70 expression on the localisation of mCherry-nCLU under conditions of stress (Appendix A). mCherry-nCLU nuclear accumulation was significantly enhanced by stress in cells co-expressing GFP (Fn/c = 0.3 vs. 0.5; *p* < 0.0001), in contrast to in cells co expressing GFP-KU70. The finding that KU70 overexpression could reduce stress-induced nuclear accumulation of nCLU suggests that in the cytoplasm, KU70 may preferentially bind IPO13 over nCLU, and, therefore, it can be speculated that under conditions of stress, nCLU likely sequesters KU70 from IPO13 in the nucleus to inhibit IPO13-dependent export of KU70 rather than nCLU sequestering IPO13 from KU70.

### 3.3. FRAP Analysis Reveals IPO13 Facilitates Nuclear Import of nCLU under Stress Conditions and Export of KU70 under Steady State Conditions Which Are Downregulated by Stress

Given the exciting result that both nCLU and KU70 appear to be newly identified stress-modulated cargoes of IPO13, we utilised our established fluorescence recovery after photobleaching (FRAP) approach to analyse nuclear import kinetics of nCLU and KU70 in real time. HeLa cells—co-transfected to express GFP-nCLU with mCherry alone and IPO13 independently expressed from pIRES-IPO13 together with mCherry—were treated with or without H_2_O_2_ for 1 h prior to live cell FRAP (Figure 2a–d). Individual cell nuclei, expressing GFP-nCLU, were photobleached using high-powered 488 nm laser and then monitored every 20 s for 8 min for the return of nuclear fluorescence (see Materials and Methods, bleached area is represented by yellow dashed line in the Pre panel). Quantitative analysis of the images (Figure 2b–d) demonstrated that only a small amount of nCLU was localised within the nuclei of untreated cells co-expressing mCherry alone, with the limited recovery observed following photobleaching, indicating that there is minimal nCLU import in the absence of stress (Frec(Fn-b); Figure 2c), consistent with steady state observations (Figure 1a,b). In contrast, H_2_O_2_-treated cells co-expressing mCherry alone (pIRES, Figure 2b–d), showed significantly higher (*p* = 0.0382) levels (twofold) in the maximal recovery of nuclear fluorescence compared with untreated cells, consistent with stress-induced import of nCLU (Figure 2c). Strikingly, this significant (*p* < 0.0007) twofold increase was comparable to that observed in cells co-expressing IPO13 in the absence of stress, consistent with the idea that IPO13 is the nuclear importer for nCLU (Figure 1a,b,f,g), with a further significant (*p* = 0.0024) twofold increase in maximal nuclear recovery above that observed in the presence of H_2_O_2_-induced stress (Figure 2c). This is consistent with the idea that IPO13 can mediate nCLU nuclear import, and this is enhanced in the presence of stress.

These results were supported by measurements of the initial rate of recovery (Frec(Fn-b)/s^−1^) of nuclear fluorescence. Cells co-expressing mCherry alone under H_2_O_2_-induced stress displayed a significant (*p* = 0.0244) increase in the initial rate at which nCLU fluorescence returned to the nucleus, compared with the untreated control (Figure 2d). Similarly, the effect of co-expression of IPO13 under untreated conditions significantly (*p* = 0.0001) increased the initial rate of nCLU import more than threefold. Notably, the initial rate of return of fluorescence to the nucleus was almost fivefold higher in cells that were H_2_O_2_-treated and overexpressing IPO13 (Figure 2d) than in the absence of stress and IPO13 overexpression, and it was significantly faster (*p* < 0.0001) than untreated cells co-expressing IPO13. The clear implication is that IPO13 imports nCLU most efficiently under stress conditions.

FRAP was similarly used to analyse the nuclear export kinetics of KU70 in real time in the presence/absence of IPO13 and/or stress. HeLa cells were co-transfected to express GFP-KU70 with DsRed2 alone or DsRed2-IPO13 and treated with or without H_2_O_2_ for 1 h prior to FRAP (Figure 2e–h). In this case, the cytoplasm of cells expressing GFP-KU70 was photobleached (represented by the blue dashed line in the Pre panel) using a high-powered laser and then monitored every 20 s for 8 min for the loss of nuclear fluorescence as an accurate measure of nuclear export. Due to the size differences between the nucleus and cytoplasm, loss of nuclear fluorescence is a more accurate measure of nuclear export than return of fluorescence to the photobleached cytoplasm, which may be due to intra-cytoplasmic movement of protein [59]; compared with untreated DsRed2 co-expressing cells, GFP-KU70 showed significant (*p* < 0.0161) fourfold loss of nuclear fluorescence in cells treated with H_2_O_2_ (maximal loss of fluorescence = −0.75 and −0.2; Figure 2g), consistent with the idea that oxidative stress reduces KU70 nuclear export, as shown in CLSM analysis (Figure 1c,d). Strikingly, in untreated cells co-expressing DsRed2-IPO13, GFP-KU70 nuclear export was significantly (*p* < 0.0161) enhanced by approximately threefold compared with cells co-expressing DsRed2 alone (Figure 2g), consistent with our previous observations that IPO13 promotes KU70 nuclear export under steady state conditions. Maximal nuclear loss of GFP-KU70 was significantly (*p* = 0.0133) lower in DsRed2-IPO13 co-expressing cells treated with H_2_O_2,_ compared with untreated cells, confirming that although IPO13 facilitates export of KU70 under basal conditions, oxidative stress reduces the efficiency.

Consistent with this idea, stress was found to significantly (*p* < 0.0496) decrease the initial rate of export of GFP-KU70 in cells co-expressing DsRed2, whereas untreated cells showed significantly (*p* < 0.0001) increased (approx. twofold) initial rate of export of KU70 in the presence of co-expressed DsRed2-IPO13 (Figure 2h). The effect of IPO13 co-expression on the initial rate of KU70 nuclear export was significantly (*p* = 0.0025) reduced (twofold) by H_2_O_2_ treatment, consistent with IPO13 being an efficient nuclear export receptor for KU70 under steady state conditions and less efficiently so in stress. Thus, oxidative stress appears to elicit a switch in transport roles, as shown here with nuclear export of KU70 inhibited in stress, while nCLU nuclear import is enhanced.

### 3.4. IPO13 Is a Key Regulator of DNA Repair

Our data demonstrates that stress inhibits IPO13-dependent nuclear export of KU70, as well as enhancing nuclear import of nCLU, which is known to favour nCLU-KU70-binding and reduces KU70-DNA-binding [36,38]. Overall, this would impact KU70′s nuclear role as part of the critical KU heterodimer that recruits NHEJ repair machinery to repair DNA double-strand breaks that are the result of oxidative stress insult [55,60,61]. To confirm a functional role for IPO13 in this process, we investigated the effect of IPO13 on both the amount and subsequent repair of H_2_O_2_-induced DNA damage. HeLa cells transfected to express GFP or GFP-IPO13 were assayed for DNA damage using an in-gel neutral comet assay (Figure 3a,b). DNA damage induced by 50 μM H_2_O_2_ was measured by comet assay either immediately following treatment or after a 2 h recovery after the medium was replaced with fresh media. Untreated cells (UT) were incubated in DMEM without H_2_O_2_ and otherwise treated identically. The percentage DNA within the comet tail was used to analyse DNA damage, whereby the greater the damage to cellular DNA, the higher the percentage of DNA located within the comet tail. Untreated cells ectopically expressing GFP-IPO13 showed significantly (*p* < 0.0001) less DNA damage (~5% DNA in comet tail, Figure 3b), but these cells were significantly (*p* < 0.0001) less resistant to stress insult, showing increased levels of DNA damage induced by H_2_O_2_ compared with GFP-expressing cells (~32% DNA in comet tail compared with 26%). Interestingly, all cells recovered to the same extent in this replicate regardless of IPO13 expression, with no significant difference in the percentage of DNA within the comet tails after 2 h (~5% DNA in comet tail), but in most other replicate experiments, GFP-IPO13-expressing cells showed significantly more DNA damage repair after 2 h than cells expressing GFP alone (data not shown), indicating some degree of variability in the extent of DNA repair induced by IPO13.

Comparable experiments were performed using HeLa cells after siRNA-induced knockdown of IPO13 (Figure 3c,d); IPO13 knockdown resulted in a significant (*p* < 0.0001) increase in the level of DNA damage in untreated cells compared with the non-targeting (NT) siRNA control (Figure 3d). Importantly, these cells were also significantly (*p* < 0.0001) less sensitive to H_2_O_2-_induced DNA damage, which is the opposite of what was observed with IPO13 overexpression.

After 2 h, these cells also had a significantly (*p* < 0.0001) greater amount of DNA damage compared with the control (NT) group. Together, these results indicate that IPO13 expression plays a significant but opposing role in the repair of/protection from basal DNA damage, where it appears to play a protective role, and in DNA damage induced by H_2_O_2_, where IPO13 appears to contribute to DNA damage/hinder DNA repair.

Since nCLU is a cargo of IPO13 during stress conditions and is known to sequester KU70 away from DNA [36,38], it seemed reasonable to hypothesise that IPO13-dependent effects on DNA damage and repair may be mediated, at least in part, through the increased nCLU localisation in stress, dependent on IPO13. To investigate this, we examined whether ectopic expression of nCLU affected the level of DNA damage induced by H_2_O_2_, at the same dosage used in the live cell experiments and the contribution, if any, of IPO13. HeLa cells with and without IPO13 knockdown were transfected to express GFP or GFP-nCLU (Figure 3e,f). When IPO13 was expressed at normal levels (NT control), GFP-nCLU expression induced a significant (*p* < 0.0001) increase in damaged DNA in H_2_O_2_-treated cells compared with cells expressing GFP alone, consistent with the role of nCLU in reducing DNA repair. This was in stark contrast to cells treated with IPO13 siRNA, where ectopic expression of nCLU had no significant effect on the damage induced by oxidative stress (~38% DNA in comet tail in both samples). These results suggest that the IPO13-dependent contribution to DNA damage during stress is mediated in large part through effects on nCLU. nCLU did not appear to play a significant role in DNA damage recovery in these experiments (Figure 3f). Given that not all cells analysed in these comet experiments were expressing the transfected constructs, and this cannot be determined by this method, it is likely that the results here represent an underestimation of the total effect of overexpression and knockdown on DNA damage.

### 3.5. IPO13 Is a Key Regulator of Apoptosis

Since IPO13 appears to impact DNA damage repair in response to oxidative stress through regulating subcellular localisation of nCLU and KU70, we hypothesised that the stress-induced cellular death (via the intrinsic apoptosis pathway also regulated by these proteins) may also be impacted by IPO13. To explore this, IPO13^+/+^ and IPO13^−/−^ knock-out embryonic stem cells (ESCs), ectopically expressing either GFP or GFP-nCLU, were treated with H_2_O_2_ in dosages ranging from sub-lethal (125 μM) to potentially lethal concentrations (300–600 μM). Cells were stained with propidium iodide (PI) prior to flow cytometric analysis to determine the proportion of dead cells within the GFP-positive population (Figure 4a,b). In the IPO13^+/+^ cell line, ectopic expression of GFP-nCLU rendered cells more sensitive to oxidative stress-induced cell death compared with GFP alone across the lethal range of stress, where at 300 μM, ectopic expression of GFP-nCLU increased death from ~35% to ~48% (*p* = 0.0089; Figure 4b), while at 600 μM, death was increased from ~43% to ~61% (*p* = 0.0001). In comparison, ectopic expression of GFP-nCLU had no significant effect on cell viability in IPO13^-/-^ cells exposed to oxidative stress at either sub-lethal or lethal levels, with all samples showing protection against stress-induced cell death compared with IPO13^+/+^ samples under the same conditions. These results clearly demonstrate that nCLU-dependent cell death induced by oxidative stress is dependent on the expression of IPO13. Further, given nCLU’s established pro-apoptotic role, this implies that IPO13 also plays a significant role in the induction of stress-induced apoptosis.

To confirm that the enhanced cell death observed in IPO13^+/+^ cells in response to oxidative stress is mediated through apoptosis, we carried out annexin V/PI double-staining in IPO13^+/+^ and IPO13^−/−^ ESC lines treated with 12 μM camptothecin (CTH, an inhibitor of topoisomerase 1 and inducer of apoptosis [62]; Figure 4c,d). IPO13 knock-out (IPO13^−/−^) significantly protected ESCs from CTH-induced apoptosis, reducing the apoptotic cell population to approximately 40% compared with IPO13^+/+^ cells, which were approximately 80% apoptotic (*p* = 0.0032) under the same conditions. Overall, these results confirm that IPO13 significantly contributes to the induction of apoptosis, and they suggest—together with the data that demonstrate IPO13 imports nCLU under stress and that nCLU-dependent cell death is dependent on the expression of IPO13 (Figure 4a,b)—that this contribution to apoptotic cell death is likely mediated, at least in part, by IPO13 through the nuclear import of nCLU under stress.

### 3.6. IPO13 Traffics Efficiently under Cellular Stress

The abundance of stress response proteins observed in the Y2H screen, coupled with the unusual nuclear import ability under stress conditions that we have identified here for IPO13 and nuclear export ability previously [29], suggests that IPO13, unlike other IPOs, can traffic actively as a nuclear transporter during stress. To begin to assess this formally, we firstly confirmed that treatment of HeLa cells with 125 μM H_2_O_2_ for 1 h was sufficient to mislocalise Ran from the nucleus to the cytoplasm (Fn/c = 1.7 and 0.4, respectively, *p* < 0.0001, (Figure 5a,b)). This was also observed in other cell lines used in this study (Appendix A) as well as after heat shock (Figure 5a,b), confirming that the conditions used throughout this study are sufficient to cause the Ran gradient collapse.

We next examined the localisation of the nuclear importers IMPα, IPOβ1, and IPO7, nuclear exporter XPO1, and bidirectional nuclear transporter IPO13, in the absence and presence of oxidative stress, using ectopically expressed GFP-tagged proteins in HeLa cells (Figure 5c–v). Digitised images were then subjected to Fn/c analysis as well as an analysis to determine the nuclear-to-nuclear-envelope-fluorescence ratio (Fn/ne) of each transport protein. Under untreated conditions, GFP-IMPα localisation was diffuse throughout the nucleus and cytoplasm (Fn/c = 1.2, Figure 5c,d). Strikingly, after stress treatment, IMPα became mostly nuclear, mislocalising considerably (Fn/c = 5.5), consistent with previous reports [10]. Additionally, Fn/ne analysis indicates that IMPα localisation at the nuclear envelope was significantly (*p* = 0.0021) reduced after stress treatment (Fn/ne = 1.2 and 0.9, respectively, where an Fn/ne value below 1 indicates accumulation at the nuclear envelope and above 1 indicates diffuse localisation throughout the nucleus and nuclear envelope) (Figure 5e,f). These drastic alterations to IMPα localisation suggest that during cell stress, IMPα is likely unable to return to the cytoplasm after an import event. In contrast, while IPOβ1 demonstrated a mostly diffuse subcellular localisation under normal conditions (Fn/c = 1, Figure 5g,h), oxidative stress induced a small but significant decrease in nuclear IPOβ1 (Fn/c = 0.9, *p* < 0.0001). More striking was the enhancement of IPOβ1 at the nuclear envelope following stress (Fn/ne = 1.3 and 1.8, *p* < 0.0001, Figure 5i,j), suggesting that the stress-induced reduction of IPOβ1 in the nucleus may be explained by accumulation of the protein at the nuclear envelope. The observed mislocalisations of both IMPα and IPOβ1 indicate that the conventional IMPα/IPOβ1 nuclear import pathways are impaired here under conditions of stress. The localisation of GFP-IPO7 was highly nuclear under untreated conditions (Fn/c = 3.3, Figure 5k,l); however, it became considerably less nuclear in stress-treated cells (Fn/c = 2.4, *p* < 0.0001). There was no significant change to the amount of GFP-IPO7 accumulated at the nuclear envelope (Figure 5m,n). GFP-XPO1 was mostly nuclear under normal conditions (Fn/c = 1.9, Figure 5o,p), with stress inducing a small but significant decrease in nuclear GFP-XPO1 (Fn/c = 1.7, *p* = 0.0257) and an increase in accumulation at the nuclear envelope (Figure 5q,r, Fne/n = 1.1 and Fne/n = 1.3, respectively, *p* < 0.0001). Finally, GFP-IPO13, which was highly nuclear under untreated conditions, remained highly nuclear under stress with no significant change in localisation between the nucleus and cytoplasm (Fn/c approx. 2.6 vs. 2.7; *p* = NS, Figure 5s,t) and no significant change in localisation at the nuclear envelope (Fne/n of 1 under both conditions) (Figure 5u,v). Taken together, this screen has identified that IPO13 appears to be resistant to stress-induced mislocalisation in stark contrast to all the other IMPs/IPOs/XPOs tested. These data, taken together with our data indicating that IPO13 facilitates nuclear import of nCLU under stress conditions (Figure 2a–d), suggest that IPO13, in direct contrast to other members of the IPO superfamily, is able to continue to traffic as a nuclear transporter under conditions of cell stress.

We confirmed these observations using FRAP, comparing IPO13 to IPO7, an IPOβ homologue, which, like IPO13, binds cargoes directly and transports them without the need for an adapter, such as IMPα. HeLa cells transfected to express GFP-tagged IPO7 or -IPO13 were examined without or with oxidative stress (125 mM H_2_O_2_) (Figure 6). After being subjected to an entire nuclear photobleach (represented by yellow dashed line in the Pre panel), GFP-IPO7 and -IPO13 both demonstrated rapid and consistent return of fluorescence to the photobleached region, indicating functional nuclear import of the IPO proteins (Figure 6a). Quantitative analysis of the images (Figure 6b–d) demonstrated that although the two IPOs recovered to approximately the same extent following nuclear photobleach (maximal recovery = 0.3 and 0.4; Figure 6c), IPO13 has a much faster rate of return (t_1/2_; Figure 6d), suggesting an innately faster cycling speed. When subjected to oxidative stress, the nuclear recovery of IPO7 was significantly impaired, recovering to only ~36% of its maximal recovery without stress (maximal recovery = 0.3 and 0.1, *p* = 0.018) and with a more than 2.4-fold decrease in the rate of transport (Figure 6d). IPO13 on the other hand showed continued efficient transport despite the stress conditions. Maximal recovery was somewhat decreased, dropping by approximately 42% compared with 64% reduction observed for IPO7 under the same conditions (Figure 6c).

In stark contrast to IPO7, the kinetics of IPO13 nuclear import was not significantly affected by the oxidative stress (t_1/2_ = 62 and 51, respectively; Figure 6d). To examine whether this is a unique response to oxidative stress or a general quality of IPO13, the experiment was repeated using sub-lethal heat shock stress (again, the treatment conditions were confirmed to induce gross changes in nuclear transport machinery, as demonstrated by nuclear depletion of Ran (Figure 5a,b and Appendix A)). Interestingly, after heat shock, IPO13 displayed no significant loss in nuclear recovery (Appendix A–d) and the kinetics of nuclear import, measured as t_1/2_, also remained constant before and after heat shock treatment (Appendix A), indicating that IPO13 was unaffected by heat shock stress. This was in drastic contrast to IPO7, where maximal recovery of nuclear fluorescence significantly decreased post heat shock, and the kinetics of nuclear import were significantly slower, as demonstrated by the significant increase in t_1/2_ (Appendix A). Taken together, these data indicate that IPO13 nuclear trafficking is largely unaffected by cellular stress in contrast with other IPO family members. Given that IPO13, unlike other IPOs tested, continues to traffic under conditions of stress that result in mislocalisation of Ran to the cytoplasm (Figure 5a,b and Appendix A), we speculate that IPO13 may be less dependent on Ran for trafficking under stress, which is consistent with its previously reported reduced dependency on Ran for nuclear export [21].

In summary, we conclude that IPO13 in contrast to other members of the IPOβ superfamily of transport proteins retains the ability to traffic as a nuclear transporter under conditions of cell stress, can mediate the stress-induced import of nCLU, and thereby plays a role in the cell death decision underlying the cellular response to stress. Import of nCLU by IPO13 is, at least in part, required for stress-induced cell death, which likely acts to antagonize the DNA repair function of KU70.

## 4. Discussion

The results in this study demonstrate that IPO13 continues to traffic as a nuclear transporter under conditions of stress, importing nCLU. This, together with our recent work [29], is the first report of an IPOβ superfamily member mediating efficient nuclear transport of a cargo under stressed conditions (as depicted in the model in Figure 7), which usually inhibit IPO-mediated nuclear transport due, in part, to the collapse of the Ran gradient. Alternative transport pathways for cellular stress response proteins that consequently must exist are largely uncharacterised, and, thus, our study provides key insight into the role of IPO13 in cellular stress signalling and fills a significant gap in the literature.

Our FRAP kinetic experiments confirm that unlike other IPOβs (including IPO7), IPO13 remains efficiently mobile between the nucleus and the cytoplasm under various cellular stress condition. Given that we observe this despite the Ran gradient collapse, this suggests that IPO13 is less dependent on Ran than other IPOs to traffic appropriately, which likely underlies this unique property of IPO13 transport during stress. To our knowledge, the Hikeshi protein, which is not a member of the IPOβ family, is the only other transporter identified thus far to traffic efficiently in stress. Hikeshi was previously identified as a nuclear import carrier for the molecular chaperone Hsp70s in response to heat shock stress, displaying Ran-independent activity [63]. However, this import activity is specific to this single import cargo and does not appear to extend to other cargoes. In response to heat shock stress, Hikeshi expression is upregulated [63]. Regulated expression of nuclear transport proteins is an efficient method of modulating the subcellular localisation of protein cargo, whereby reducing or enhancing the nuclear transport receptor expression can, in turn, respectively downregulate or promote the transport of cargo proteins [22]. In the case of IPO13, we do not observe changes in expression of IPO13 in the presence of, or during recovery from, cellular stress (Appendix A), and so it is unlikely that altered IPO13 expression in response to stress is responsible for import of nCLU or other cargo by IPO13 under stress. Thus, it is probable that other mechanisms regulate IPO13-mediated nuclear import under these conditions. One such mechanism may be the loss of competition with other IPOs. For example, it is established that the glucocorticoid receptor GR is a cargo of multiple IPOs, including IPOα/IPOβ, IPO7, and IPO13 [27,64]. It is also known that GR undergoes hormone-independent import into the nucleus under conditions of stress [65]. However, under conditions of stress, consistent with our observations here of reduced IPO7 transport efficiency under stress, and mislocalisation of all three, IMPα/IPOβ- and IPO7-mediated nuclear import are reported to be impaired [10,66]. Therefore, IPO13 may function as an alternative nuclear transporter when other transport pathways are inhibited, thereby guaranteeing the nuclear transport of critical signalling proteins to manage and enable the cellular response to stress. Our observed minimal nCLU nuclear import under normal conditions, despite a similar binding affinity (as determined by IP and AlphaScreen), suggests that this mechanism could be at play in our study.

Another potential mechanism of regulating stress-induced transport is the modulation of protein function and/or protein–protein interactions by post translational modification (PTM). nCLU has been observed to be a non-glycosylated protein in the cytoplasm and a glycosylated protein within the nucleus [67]. Although the significance of the glycosylation to the nuclear import of nCLU has not been elucidated, use of glycosylation to regulate nuclear transport of cytosolic proteins is not unusual, as seen, for example, with the stress responsive transcription factor NF-κB [68,69]. Clearly, the contribution of nCLU glycosylation to nCLU nuclear import and similar stress-induced PTMs to other IPO13 cargo merits further investigation. Finally, while the expression of IPO13 does not appear to change with stress, differential expression of the cargo itself under such conditions is a possible mechanism of regulation of cargo nuclear import. The transcriptional activity of Hypoxia-inducible factor1-α (HIF1-α) is stimulated in response to stress and, in turn, promotes the expression of nCLU under these conditions [58,70]. Clearly, expression of nCLU is likely to potentiate nCLU import by IPO13.

Through our cell-based screen, we confirmed two new transport cargoes of IPO13, nCLU, and KU70. Recently, using a SILAC/mass spectrometry approach, Baade et al. confirmed IPO13 interaction with KU70 and further confirmed that this was enhanced in the presence of RanQ69L, a Ran mutant that cannot be hydrolysed by RanGAP. Clearly, these data support the finding that IPO13 is a nuclear exporter for KU70. In contrast to the stress-induced import of nCLU, the IPO13-dependent export of KU70 appears to be inhibited by oxidative stress (see Figure 7). It is established that protein cargoes can become mislocalised under such conditions, due to the inhibition of their nuclear transport. However, given that IPO13 mobility and transport capability is not significantly affected by stress, it seems likely that KU70 accumulation in the nucleus is through sequestration away from IPO13 to prevent its nuclear export. Under conditions of stress, KU70 is thought to be bound by nCLU within the nuclear compartment, with CLU overexpression resulting in reduction of KU70 DNA end-binding [37,38]. Additionally, nuclear localisation of nCLU was shown to coincide with DNA fragmentation. The implication is that nCLU may sequester KU70 from roles in DNA repair under these conditions [71]. Given that nCLU is thought to bind to KU70 at C-terminal residues also important for KU70 binding to DNA, this may suggest that nCLU modulates KU70 DNA end-binding by competing for KU70′s DNA-binding domain [38]. Importantly, our Y2H screen indicated that IPO13 bound to residues 477–609 of the KU70 C-terminus (Table 1). If IPO13 and nCLU bind to the same sites on KU70, this could explain why we observe that under stress, where nCLU localises in the nucleus, where it is known to bind to KU70, binding of KU70-IPO13 and subsequent export of KU70 is downregulated (Figure 1k and Figure 2e–h). Further, the suggested amino acid residues minimally required of nCLU for association with KU70 (326–365) [36] also overlap with those found to interact with IPO13 in our Y2H analyses (248–348), suggesting that when IPO13 imports nCLU, the affinity of the interaction between nCLU and KU70 may outcompete the binding of IPO13 to nCLU and drive the dissociation of the import complex, despite the stress-induced loss of nuclear Ran. This could also explain the downregulation of KU70 nuclear export. Such a mechanism is supported by our IPs (Figure 2n,o), where ectopic expression of nCLU can reduce the amount of IPO13 co-precipitated with KU70. Such a release mechanism would not be unexpected given previous findings that IPO13 is atypical in its release of the export cargo eIF1A, which must also be competitively displaced by import cargo in the cytoplasm [21,24].

Although it remains unclear whether sequestering KU70 from DNA repair is responsible for nCLU-mediated apoptosis, it would appear that at least the binding of nCLU to KU70 under these conditions is crucial; stable expression of the GFP-tagged nCLU C-terminal minimal KU70-binding domain in MCF-7 cells has been shown to be sufficient to induce cell death to the same extent as the full length GFP-tagged nCLU fusion protein, and mutation of the KU70-binding domain within nCLU is reported to prevent nCLU-induced apoptosis whilst retaining nuclear localisation capability [36]. Our comet assay is consistent with this report, suggesting that IPO13 contributes to DNA damage during stress (Figure 3a–d) and that GFP-nCLU contributes to DNA damage during stress in an IPO13-dependent manner (Figure 3e,f), suggesting that the import of nCLU by IPO13 is interfering with the DNA repair activity of the cell. Additionally, under stress, IPO13 imports nCLU into the nucleus, where it carries out a pro-apoptotic role, which appears to be significantly dependent on IPO13 (Figure 4).

Although IPO13 appears to facilitate cell death via apoptosis under certain stresses and its knock-out decreases the sensitivity of cells to stress induced death, we also found that IPO13 appears to contribute to DNA repair in untreated and recovering cells, as seen in the comet assays (Figure 3a–d). Notably, KU70 also plays an anti-apoptotic role within the cytoplasm, where it binds to and sequesters Bax from the mitochondria [53]. Given that nCLU-mediated apoptosis has been shown to be Bax-dependent [54], it seems plausible that under normal cellular conditions, IPO13 functions to export KU70 to the cytoplasm to maintain the cytoplasmic pool of KU70 required to take part in this important anti-apoptotic role (see Figure 7). The role of IPO13 in the decision between promoting life and death is, therefore, likely to be complex and dependent on the level and duration of the stress agent. Whilst IPO13 may initially enable cellular stress pathways geared at promoting survival, if the stress is unresolved, IPO13 may then switch to promoting the elimination of damaged cells. Future investigation into the role of IPO13 in the cellular response to stress will benefit from detailed study into the contribution of IPO13 to a variety of cellular stress response pathways under varying degrees of stress, aided by comparison of the IPO13 interactome between steady state and stress, which is the focus of future work in this laboratory and will likely unravel the delicate balance provided by this key regulator of the cell-fate decision.

## Figures and Tables

**Figure 1 cells-12-00279-f001:**
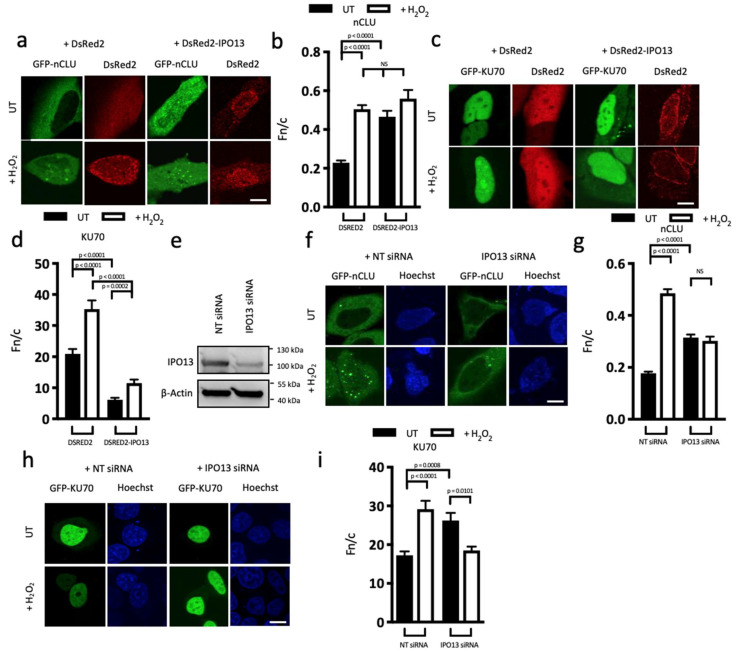
IPO13, respectively, is an import receptor for nCLU and export receptor for KU70. (**a**–**d**) HeLa cells were subjected to CLSM 16 h post-transfection to co-express either DsRed2 or DsRed2-IPO13 with GFP-nCLU, Scale bar = 10 μM (**a**) or GFP-KU70, Scale bar = 10 μM (**c**) and treated ± 125 μM H_2_O_2_ for 1 h prior to imaging live. Quantitative analysis of GFP-nCLU (**b**) or GFP-KU70 (**d**) localisation was carried out using the ImageJ software on images, such as those in (**a**,**c**)**,** to determine the nuclear-to-cytoplasmic-fluorescence ratio (Fn/c), as described in Materials and Methods. Values represent the mean ± SEM (n > 50 cells) from a single typical experiment from a series of 2 (**b**) or 3 (**d**) similar experiments. (**e**–**i**) HeLa cells were subjected to CLSM 72 h post-transfection with non-targeting or IPO13 siRNA. (**e**) Total cell extracts were probed by Western blotting using rabbit-anti-IPO13 (Protein Tech), with mouse-anti-actin (Abcam, Cambridge, UK) as a control and imaged using the ChemiDoc Gel Imaging System (Biorad, Hercules, CA, USA). At 16 h post-transfection, cells were transfected with either GFP-nCLU, Scale bar = 10 μM (**h**) or GFP-KU70, Scale bar = 10 μM (**h**) and treated with H_2_O_2_ as per (**a**,**c**) above. Quantitative analysis of GFP-nCLU (**g**) or GFP-KU70 (**i**) localisation was carried out as in (**b**,**d**). Values represent the mean ± SEM (n > 31 cells) from a single typical experiment from a series of 2 (**g**) or 3 (**i**) similar experiments. (**j**) HeLa cells were transfected to co-express GFP or GFP-nCLU and dsRed2-IPO13. At 16 h post-transfection, cells were treated with 125 μM H_2_O_2_ for 1 h before lysis and immunoprecipitation (IP) using GFP-Trap beads (Chromotek, Proteintech, Rosemont, IL, USA). Input or IP samples were probed by Western Blotting using rabbit-anti-IPO13 (Aviva, Protein Tech) or mouse-anti-GFP (Roche, Basel, Switzerland) antibodies. (**k**) HeLa cells were transfected to express GFP-IPO13. At 16 h post-transfection, cells were treated with 125 μM H_2_O_2_ for 1 h, after which an additional glutaraldehyde–protein crosslinking step was performed (see Materials and Methods) before lysis and IP, as in (**j**). Input or IP samples were probed by Western Blotting using mouse-anti-KU70 (Abcam) or mouse-anti-GFP (Roche) antibodies. The amount of 30 nM of His_6_-tagged GFP-nCLU (**l**) or -KU70 (**m**) was incubated with increasing concentrations of biotinylated-GST or biotinylated-GST-IPO13 prior to AlphaScreen analysis to determine the K_D_ of the interaction. Assays were performed in triplicate; graphs show mean AlphaScreen counts ± S.E.M (n = 3) from a single typical experiment, representative of two independent assays. (**n**,**o**) HeLa cells were transfected to co-express GFP or GFP-KU70 and mCherry or mCherry-nCLU. At 16 h post-transfection, cells were treated with 125 μM H_2_O_2_ for 1 h, after which an additional glutaraldehyde–protein crosslinking step was performed (see Materials and Methods) before lysis and IP using GFP-Trap beads (Chromotek). (**n**) Input or IP samples were probed by Western Blotting using rabbit-anti-IPO13 (Protein Tech) or mouse-anti-GFP (Roche) antibodies. (**o**) Densitometric analysis was performed on images, such as those in (**a**), for binding of IPO13 to GFP-KU70 under H_2_O_2_-treated conditions and untreated conditions with and without co-transfection of mCherry-nCLU. The amount of co-immunoprecipitated IPO13 was normalized to the amount of immunoprecipitated GFP-KU70. Pooled results (n ≥ 4) represent the mean ± SEM for KU70-bound IPO13 in mCherry-nCLU co-transfected cells relative to mCherry co-transfected cells under untreated conditions or KU70-bound IPO13 in mCherry-nCLU co-transfected cells relative to mCherry co-transfected cells under H_2_O_2_ treated cells.

**Figure 2 cells-12-00279-f002:**
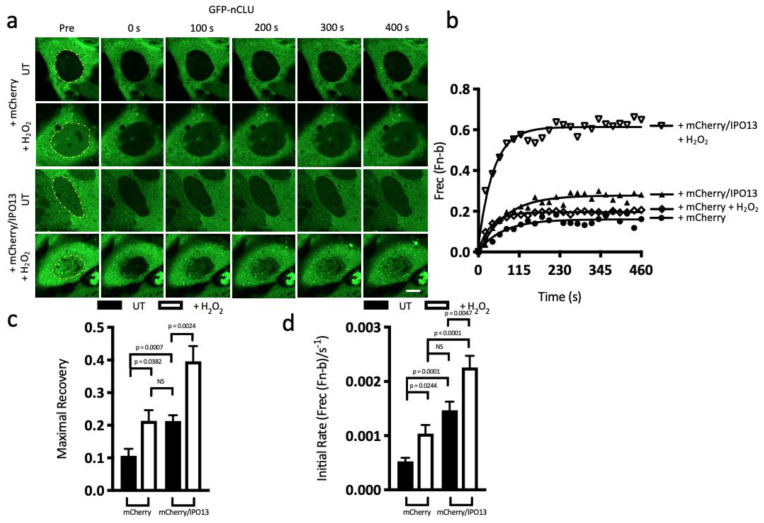
IPO13 efficiently traffics nCLU into the nucleus under oxidative stress, but IPO13-mediated nuclear export of KU70 is inhibited, as confirmed by fluorescence recovery after photo bleaching (FRAP) analysis. (**a**) CLSM images of HeLa cells transfected to co-express either mCherry or mCherry and IPO13 (expressed separately from the same plasmid using an IRES translation initiation site, pIRES) with GFP-nCLU and treated ± 125 μM H_2_O_2_ for 1 h, Scale bar = 10 μM. Cells were imaged prior to photobleaching (Pre) in the indicated nuclear region (dotted outline in yellow) and then monitored every 20 s for 8 min. (**b**) Digitised images, such as those in (**a**), were analysed to determine the fractional recovery of nuclear fluorescence (Frec(Fn-b)). Results shown are for a single representative cell under each condition. Curves, such as those generated in (**b**), were used to determine the maximal recovery of nuclear fluorescence (**c**) and the initial rate of recovery, up to 100 s post-bleaching (Frec (Fn-b)/s^−1^); (**d**) Results represent the mean ± SEM (n > 20), typical results from 3 separate experiments. *p*-values represent statistical differences as determined by Student’s *t*-test. (**e**) CLSM images of HeLa cells transfected to co-express either DsRed2 or DsRed2-IPO13 with GFP-KU70 and treated ± 125 μM H_2_O_2_ for 1 h, Scale bar = 10 μM. Cells were imaged live prior to photo bleaching (Pre) in the indicated cytoplasmic region (dotted outline in blue) and then monitored every 20 s for 8 min. (**f**) Digitised images, such as those in (**e**), were analysed to determine the fractional change of nuclear fluorescence (Frec(Fn-b)). The more negative a value in this assay, the more nuclear export is occurring; therefore, when there is less loss of nuclear fluorescence, this is indicative of less export and vice versa. Results shown are for a single representative cell under each condition. Curves such as those generated in (**b**), were used to determine the maximal loss of nuclear fluorescence (**g**) and the initial rate of export, up to 100 s post-bleaching (Frec (Fn-b)/s^−1^); (**h**) Results represent the mean ± SEM (n > 20), typical results from 3 separate experiments. *p*-values represent statistical differences as determined by Student’s *t*-test.

**Figure 3 cells-12-00279-f003:**
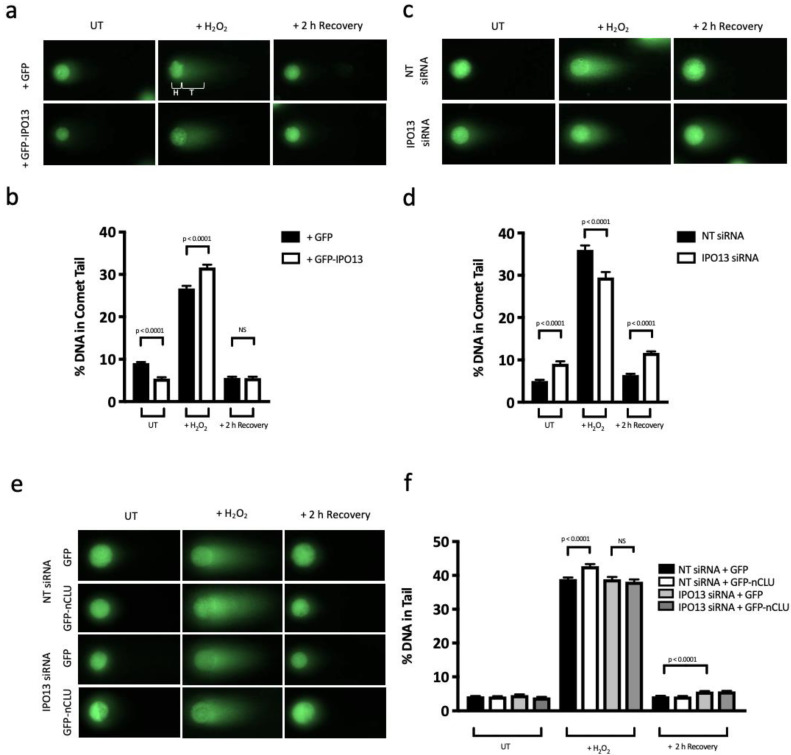
IPO13 plays a significant role in stress-induced DNA damage and repair, in part through effects on nCLU. (**a**) Fluorescence images of comets produced by in-gel neutral comet assay (single-cell electrophoresis) from HeLa cells ectopically expressing either GFP or GFP-tagged IPO13 after treatment without or with 50 μM H_2_O_2_ for 1 h or treatment followed by 2 h recovery in fresh media. H denotes the comet head and T denotes the comet tail (middle top panel). (**b**) Tail DNA content (%) was quantified using the OpenComet plugin for ImageJ to determine the percentage of DNA in the comet tail (mean ± SEM, n > 100 comets per sample). Results represent a single typical experiment from a series of 3 independent experiments. (**c**) Fluorescent images of comets produced as in (**a**) from HeLa cells transfected with either non-targeting (NT) or IPO13 siRNA after treatment as in (**a**) with 125 μM of H_2_O_2_. (**d**) Tail DNA content of pictures, such as those in (**c**), was quantified as in (**b**). (**e**) Fluorescent images of comets produced as in (**a**) from HeLa cells transfected with either NT or IPO13 siRNA that ectopically expressed either GFP or GFP-nCLU after treatment as in (**a**) with 125 μM H_2_O_2_. (**f**) Tail DNA content from images, such as those in (**e**), was quantified as in (**b**), with results representing a single typical experiment from 2 independent experiments (mean ± SEM, n > 200 comets per sample).

**Figure 4 cells-12-00279-f004:**
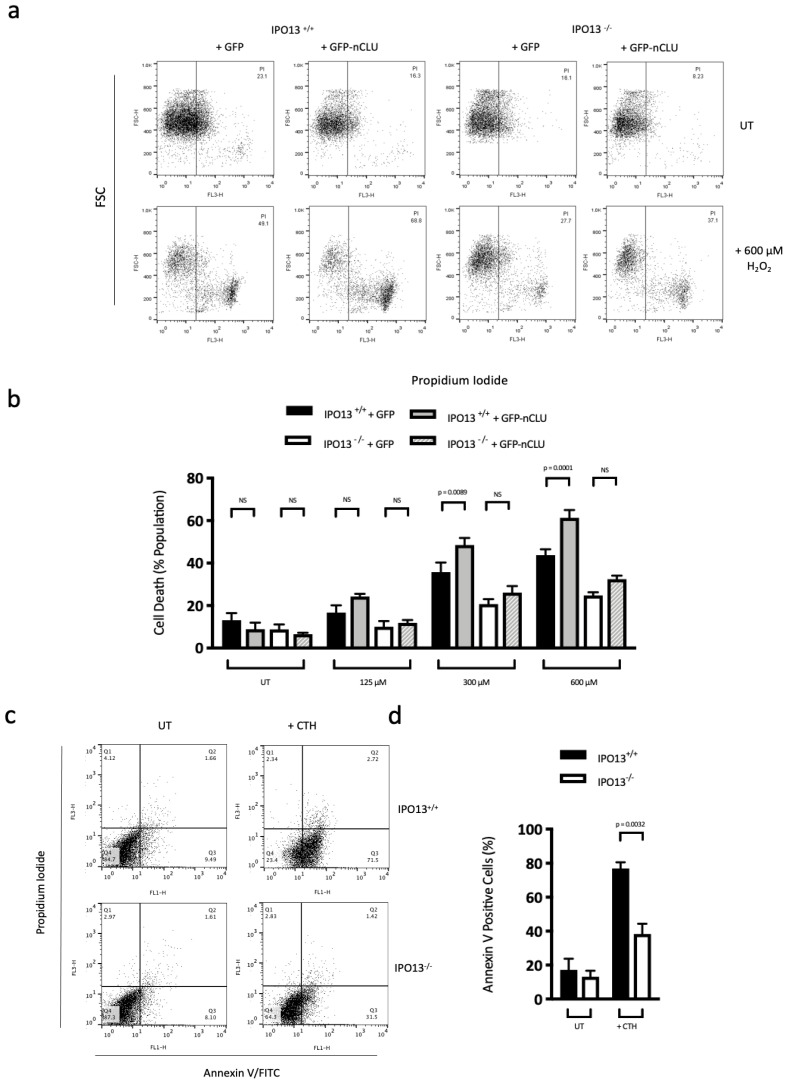
IPO13 contributes to nCLU-induced cell death and apoptosis. (**a**,**b**) Flow cytometric analysis for cell death in IPO13^+/+^ and IPO13^−/−^ ESC transfected with GFP or GFP-nCLU and treated with H_2_O_2_ for 1 h prior to FACS analysis for percentage of cell death (PI-positive cells) within the GFP- or GFP-nCLU-transfected cell populations. (**a**) Representative plots of untreated and 600 μM H_2_O_2_-treated IPO13^+/+^ and IPO13^−/−^ ESCs, gated to include the GFP positive populations. (**b**) Pooled data (n = 6 independent experiments) for % of GFP- or GFP-nCLU-expressing cells that are PI-positive (mean ± SEM) under increasing concentrations of H_2_O_2_ treatment as indicated. *p*-values represent statistical differences as determined by two-way ANOVA using Prism 7. (**c**,**d**) Flow cytometric analysis for apoptosis (Annexin V and/or PI staining) in IPO13^+/+^ and IPO13^−/−^ ESCs treated with or without 12 μM Camptothecin (CTH) for 6 h. (**c**) Representative dot plots for untreated and CTH-treated conditions are typical of three independent assays. In each panel, the upper left quadrant (Q1) shows only PI-positive cells, which are necrotic. The upper right quadrant (Q2) shows cells positive for both PI and Annexin V cells. The bottom right quadrant (Q3) shows cells positive for Annexin V only, and the bottom left quadrant (Q4) shows unstained cells. (**d**) Pooled data (n = 3 independent experiments) for % of untreated or CTH-treated IPO13^+/+^ and IPO13^−/−^ ESCs positive for Annexin V staining (Q2 + Q3) (mean ± SEM).

**Figure 5 cells-12-00279-f005:**
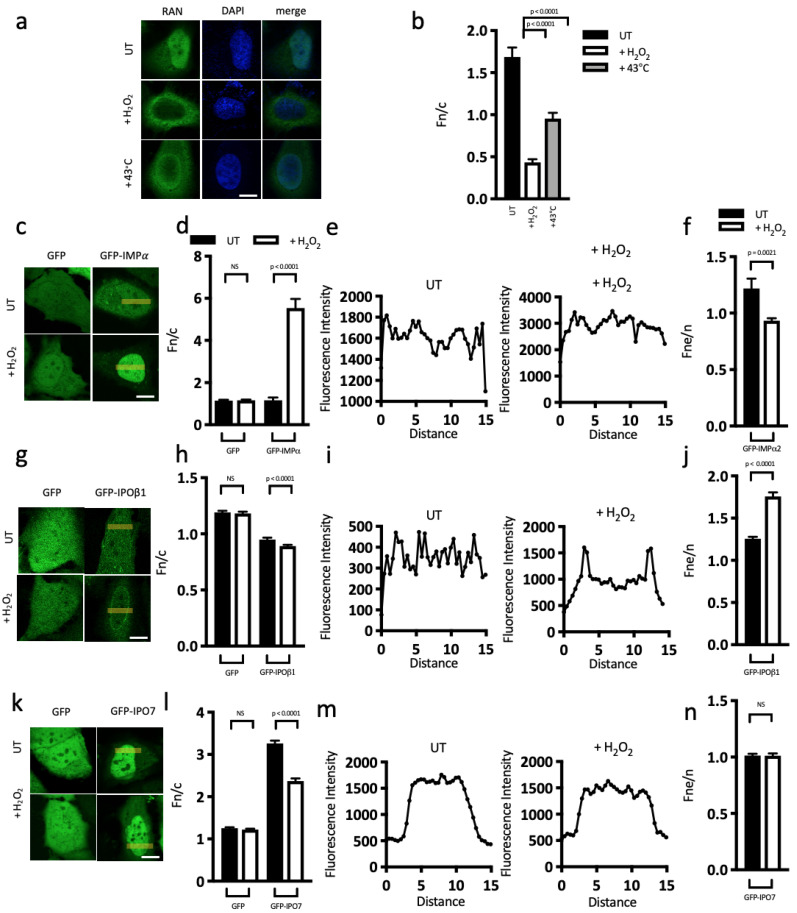
Stress disrupts the localisation of nuclear transport machinery components but not IPO13. (**a**,**b**) HeLa cells were treated ± 125 μM H_2_O_2_ for 1 h or ± 43 °C for 1 h prior to staining with mouse-anti-Ran (BD Biosciences) and counter-staining with DAPI, Scale bar = 10 μM. Quantitative analysis of Ran localisation (**b**) was carried out using the ImageJ software on images, such as those in (**a**), to determine the nuclear-to-cytoplasmic-fluorescence ratio (Fn/c) of Ran, as described in Materials and Methods. Values represent the mean ± SEM (n > 50 cells) from a single typical experiment from a series of 2 similar experiments. (**c**,**d**) Typical CLSM images of HeLa cells, 16 h post-transfection to express GFP or GFP-IMPα (**c**) and treated ± 125 μM H_2_O_2_ for 1 h prior to imaging live, Scale bar = 10 μM. Quantitative analysis of GFP or GFP- IMPα (**d**) was carried out as in (**b**). Values represent the mean ± SEM (n > 30 cells) from a single typical experiment from a series of 3 similar experiments. (**e**,**f**) Typical line fluorescence intensity histograms of GFP-IMPα (**e**) were measured across the nuclear envelope as indicated by the yellow line on the corresponding cell images (**c**). (**f**) Quantitative analysis of GFP-IMPα was carried out using the ImageJ software on images, such as those in (**c**), to determine the nuclear-envelope-to-nuclear ratio (Fne/n), as described in Materials and Methods. Values represent the mean ± SEM (n > 30 cells) from a single typical experiment from a series of 3 similar experiments. (**g**,**h**) Typical CLSM images of HeLa cells, 16 h post-transfection to express GFP or GFP-IPOβ1 (**c)** and treated ± 125 μM H_2_O_2_ for 1 h prior to imaging live, Scale bar = 10 μM. Quantitative analysis of GFP or GFP-IPOβ1 (**d**) was carried out as in (**b**). Values represent the mean + SEM (n > 30 cells) from a single typical experiment from a series of 3 similar experiments. (**i**,**j**) Typical line fluorescence intensity histograms of GFP-IPOβ1 (**i**) were measured across the nuclear envelope, as indicated by the yellow line on the corresponding cell images (**g**). (**j**) Quantitative analysis of GFP-IPOβ1 was carried out as in (**f)**. Values represent the mean ± SEM (n > 30 cells) from a single typical experiment from a series of 3 similar experiments. (**k**,**l**) Typical CLSM images of HeLa cells, 16 h post-transfection to express GFP or GFP-IPO7 (**k**) and treated ± 125 μM H_2_O_2_ for 1 h prior to imaging live, Scale bar = 10 μM. Quantitative analysis of GFP or GFP-IPO7 (**l**) was carried out as in (**b)**). Values represent the mean ± SEM (n > 30 cells) from a single typical experiment from a series of 3 similar experiments. (**m**,**n**) Typical line fluorescence intensity histograms of GFP-IPO7 (**m**) were measured across the nuclear envelope, as indicated by the yellow line on the corresponding cell images (**k**). (**n**) Quantitative analysis of GFP-IPO7 was carried out as in (**f**). Values represent the mean ± SEM (n > 30 cells) from a single typical experiment from a series of 3 similar experiments. (**o**,**p**) Typical CLSM images of HeLa cells, 16 h post-transfection to express GFP or GFP-XPO1 (**o**) and treated ± 125 μM H_2_O_2_ for 1 h prior to imaging live, Scale bar = 10 μM. Quantitative analysis of GFP or GFP-XPO1 (**p**) was carried out as in (**b**). Values represent the mean ± SEM (n > 30 cells) from a single typical experiment from a series of 3 similar experiments. (**q**,**r**) Typical line fluorescence intensity histograms of GFP-XPO1 (**q**) were measured across the nuclear envelope, as indicated by the yellow line on the corresponding cell images (**o**). (**r**) Quantitative analysis of GFP-XPO1 was carried out as in (**f)**. Values represent the mean ± SEM (n > 30 cells) from a single typical experiment from a series of 3 similar experiments. (**s**,**t**) Typical CLSM images of HeLa cells, 16 h post-transfection to express GFP or GFP-IPO13 (**s**) and treated ± 125 μM H_2_O_2_ for 1 h prior to imaging live, Scale bar = 10 μM. Quantitative analysis of GFP or GFP-IPO13 (**t**) was carried out as in (**b**). Values represent the mean ± SEM (n > 30 cells) from a single typical experiment from a series of 3 similar experiments. (**u**,**v**) Typical line fluorescence intensity histograms of GFP-IPO13 (**u**) were measured across the nuclear envelope, as indicated by the yellow line on the corresponding cell images (**s**). (**v**) Quantitative analysis of GFP-IPO13 was carried out as in (**f**). Values represent the mean ± SEM (n > 30 cells) from a single typical experiment from a series of 3 similar experiments.

**Figure 6 cells-12-00279-f006:**
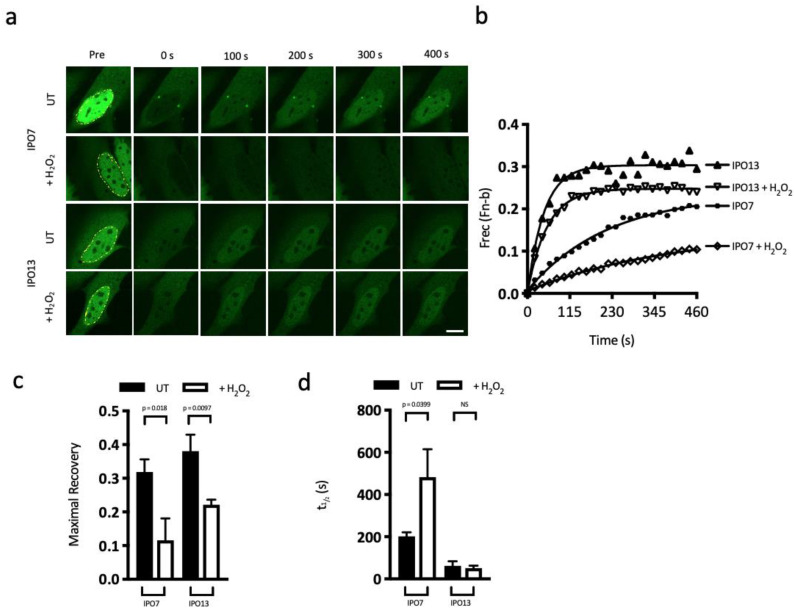
IPO13, unlike IPO7, continues to traffic into the nucleus under H_2_O_2_-induced oxidative stress. (**a**) CLSM images of HeLa cells transfected to express GFP-IPO7 or GFP-IPO13 and treated ± 125 μM H_2_O_2_ for 1 h. Cells were imaged prior to photobleaching (Pre) in the indicated nuclear region (dotted outline in yellow) and then monitored every 20 s for 8 min. (**b**) Digitised images, such as those in (**a)** were analysed to determine the fractional recovery of nuclear fluorescence (Frec(Fn-b)), Scale bar = 10 μM. Results shown are for a single representative cell under each condition. Curves, such as those generated in (**b**), were used to determine the maximal recovery of nuclear fluorescence (**c**) and the time post-bleaching to reach half-maximal recovery (t_1/2_). (**d**) Results represent the mean ± SEM (n = 20); typical results from 2 separate experiments. *p*-values indicate statistical differences as determined by Student’s *t*-test.

**Figure 7 cells-12-00279-f007:**
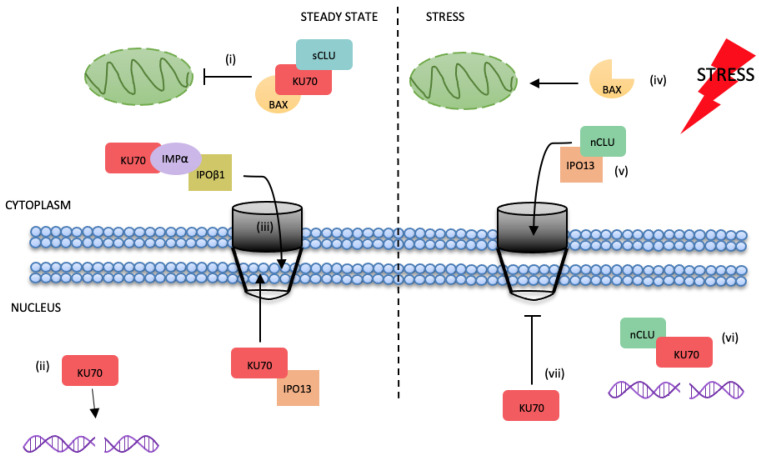
Model of IPO13s contribution to the DNA repair/Apoptosis decision axis as regulated by KU70 and CLU isoforms. (**i**) Under steady state conditions in the cytoplasm, KU70 sequesters Bax from the mitochondrion to prevent its activation of apoptosis. sCLU further stabilises the KU70/Bax binding and, thus, contributes to inhibition of Bax-mediated apoptosis. (**ii**) In the nucleus, KU70 senses DNA double-strand breaks (DSB), recruiting non-homologous end-joining (NHEJ) DNA repair machinery to sites of DSBs, resulting in damage repair. (**iii**) Nucleocytoplasmic transport facilitates KU70′s separate roles in the cell. KU70 nuclear import is facilitated by the IMPα/IPOβ1 heterodimer. The export of KU70 is facilitated by IPO13. (**iv**) In response to cellular stress, Bax is liberated from Ku70 and presumably CLU can then initiate apoptosis. (**v**) nCLU is newly expressed under these conditions and undergoes import into the nucleus by an IPO13-facilitated pathway. (**vi**) In the nucleus, nCLU binds to KU70. Under these conditions, KU70 binding to DNA is significantly reduced and (**vii**) KU70 export by IPO13 is inhibited, which both contribute to activation of Bax and shut down of DNA repair in preparation of apoptosis.

**Table 1 cells-12-00279-t001:** IPO13 interacting proteins identified by yeast-2-hybrid screen of a human testis, lung, and brain cDNA libraries using mouse IPO13 as a bait. * Predicted Biological Score (PBS Score) indicates the confidence of interaction (A being highest)—the highest PBS score obtained for each protein is given, and the screens in which the interaction was detected (at any level) are given in parentheses; # Selective interactive domain (SID) is the minimal sequence shared by all fragments of the respective protein that interacts with IPO13. N indicates proteins for which only non-sense transcripts were identified, and, as such, a SID could not be calculated. Pathways analysis was performed using the DAVID Bioinformatics Resources 6.8, using the Functional Annotation Tool for 59 of 62 IPO13 interactors that were annotated as described in Materials and Methods. Proteins annotated to stress response are blocked in red. Proteins annotated to nuclear transport are indicated by NT in the functional role column.

Prey Protein	PBS Score *	SID #	Previously Identified	Subcellular Localisation; Functional Role
Zinc Finger AN1-Type-Containing 4 (ZFAND4;ANUBL1)	A (testis)	28–119		Cytoplasm/nucleus; contains an AN1-type zinc finger and one ubiquitin-like domain/paralog of ZFAND5, which is involved in proteasome regulation/NFκB regulation
HIRA-Interacting Protein 3 (HIRIP3)	A (lung, testis)	400–523	Mackmull et al., 2017	Nucleus; may function in chromatin and histone metabolism
Kinesin Family Member 5C (KIF5C)	A (brain)	805–931	Mackmull et al., 2017	Cytoplasm/nucleus; microtubule motor protein, which may play a role in organelle transport
Protein Phosphatase 2 Regulatory Subunit B’Alpha (PPP2R5A)	A (brain, lung, testis)	143–371		Nucleus/cytoplasm; regulatory subunit of the phosphatase 2A serine/threonine phosphatase implicated in the negative control of cell growth and division
GLI-Kruppel Zinc Finger Family Member (HKR1)	A (brain, lung, testis)	237–410		Nucleus/cytoplasm/Golgi/mitochondria; may be involved in transcriptional regulation
Ribosomal Protein L31 (RPL31)	A (brain, testis)	22–110	Baade et al., 2018, Kimura et al., 2017	Cytoplasm/EC; component of the 60 s ribosomal subunit
Ran-Binding Protein 2 (RanBP2; Nup358)	B (brain, lung, testis)	1164–1489, 2890–3124		Cytoplasmic filaments of the NPC; E3 ligase, which facilitates sumoylation of certain proteins/nuclear transport. NT.
RANBP2-Like and GRIP Domain-Containing 8 (RGPD8)	B (testis)	1335–1459		NPC; may be involved in nuclear transport, particularly of RNA.
RNA Polymerase II Subunit J2 (POLR2J2)	B (lung, testis)	6–98		Nucleus; component of the RNA polymerase II transcription machinery
MGA, MAX Dimerization Protein (MGA)	B (brain, lung)	744–978		Nucleus/cytoplasm; dual-specificity transcription factor regulating the expression of both MAX-network and T-box family target genes
RANBP2-Like and GRIP Domain-Containing 6 (RGPD6)	B (brain, lung)	890–1208		NPC; thought to play roles similar to RanBP2 due to homology/nuclear transport
Zinc Finger Protein 106 (ZNF106)	B (brain, lung)	145–392		Cytoplasm/plasma membrane/nucleolus; may play a role in transcriptional regulation
ADP Ribosylation Factor-Like GTPase 6 Interacting Protein 4 (ARL6IP4)	B (brain)	179–241	Kimura et al., 2017	Nucleus/mitochondria; modulates alternative pre-mRNA splicing
Ran-Binding Protein 1 (RanBP1)	B (brain)	1–158	Mackmull et al., 2017, Baade et al., 2018	Nucleus/cytoplasm/cytoskeleton; regulates nuclear transport by increasing GTP hydrolysis induced by the Ran GTPase-activating protein RANGAP1. NT.
Signal Recognition Particle 68 (SRP68)	B (brain)	30–289		Cytoplasm/ER/nucleus/mitochondria; subunit of the signal recognition particle, which transports secreted and membrane proteins to the endoplasmic reticulum for processing
Clusterin (CLU)	B (brain, testis)	248–348		Extracellular/mitochondria/cytoplasm/ER/nucleus; suggested to be involved in several basic biological events, such as cell death, tumor progression, and neurodegenerative disorders. NT.
Protein Phosphatase, Mg2+/Mn2+-Dependent 1E (PPM1E)	C (brain)	557–755		Nucleus/cytoplasm/mitochondria; serine/threonine phosphatase
Activity-Dependent Neuroprotector Homeobox (ADNP)	D (testis, brain)	206–500		Nucleus/cytoskeleton/cytoplasm/EC; upregulated by vasoactive intestinal peptide and may be involved in its stimulatory effect on certain tumor cells/predicted transcription factor
Amyotrophic Lateral Sclerosis 2 Chromosome Region Candidate 11 (ALS2CR11)	D (testis)	3–337		Nucleus/cytoplasm; contains a calcium-dependent membrane-targeting C2 domain, often found in proteins involved in membrane trafficking and signal transduction
Chromosome 11 Open Reading Frame 63 (C11orf63)	D (testis)	175–546		Nucleus/cytoplasm/Golgi/mitochondria; unknown
KAT8 Regulatory NSL Complex Subunit 1 (KANSL1; KIAA1267)	D (brain, lung, testis)	514–712		Nucleus/cytoplasm/cytoskeleton; involved in histone acetylation
LCA5L, Lebercilin-Like (LCA5L)	D (testis)	14–195		Nucleus/cytoplasm/cilia; ciliary protein, precise function unknown
Nucleoporin 153 (Nup153)	D (lung, testis)	182–496		NE/nucleus/cytoplasm; component of the NPC/contains an RNA-binding domain. NT.
Splicing Factor 3b Subunit 2 (SF3B2)	D (testis)	492–719	Mackmull et al., 2017, Baade et al., 2018	Nucleus/cytoplasm; RNA splicing
Zinc Finger Protein 479 (ZNF479)	D (testis)	1–211		Nucleus/cytoplasm; may be involved in transcriptional regulation
Ankyrin Repeat and KH Domain-Containing 1 (ANKHD1)	D (lung)	1549–1999		Nucleus; scaffolding protein, which may play a role in apoptosis regulation
Histone Deacetylase 9 (HDAC9)	D (lung)	191–460		Nucleus; histone deacetylase
Cell-Cycle-Associated Protein 1 (CAPRIN1; M11S1)	D (lung)	341–694		Cytoplasm/plasma membrane/nucleus; may regulate the transport and translation of mRNAs involved in cell proliferation and migration
MDS1 And EVI1 Complex Locus (MECOM)	D (lung)	681–968		Cytoplasm/nucleus/Golgi/EC; transcriptional regulator and oncoprotein that may be involved in hematopoiesis, apoptosis, development, and cell differentiation and proliferation
Nucleoporin 58 (NUP58; NUPL1)	D (lung)	33–372		NPC; component of the NPC. NT.
Proteoglycan 4 (PRG4)	D (lung)	890–1066		Extracellular, nucleus, cytoplasm; plays a role in boundary lubrication within articulating joints
Retinoic Acid Receptor Gamma (RARG)	D (lung)	72–270		Nucleus; ligand-dependent transcriptional regulator
Regulatory Factor X7 (RFX7; RFXDC2)	D (lung)	100–475		Nucleus/cytoplasm; transcription factor
Zinc Finger Protein 280D (ZFN280D; SUHW4)	D (lung)	52–394		Nucleus/cytoskeleton/cytoplasm/Golgi; may function as a transcription factor
Zinc Fingers and Homeoboxes 3 (ZHX3)	D (lung)	292–627	Mackmull et al., 2017	Nucleus/mitochondria; may function as a transcriptional repressor in conjunction with the transcription factor NFY-A
ArfGAP with RhoGAP Domain, Ankyrin Repeat and PH Domain 2 (ARAP2)	D (brain)	123–618		Cytoplasm/nucleus/cytoskeleton; modulates actin cytoskeleton remodelling/plays a role in the regulation of focal adhesion dynamics
ADP Ribosylation Factor 1 (ARF1)	D (brain)	1–130	Baade et al., 2018	Cytoplasm/endosome/extracellular/Golgi/plasma membrane/cytoskeleton/ER/mitochondria/peroxisome/vacuole; ADP-ribosyltransferase involved in protein trafficking among different compartments
Coronin 2B (CORO2B)	D (brain)	186–312		Cytoskeleton/cytoplasm/nucleus/plasma membrane; may play a role in the reorganization of neuronal actin structure
Filamin B (FLNB)	D (brain)	2499–2578	Baade et al., 2018, Kimura et al., 2017	Cytoskeleton/cytoplasm/EC/plasma membrane/nucleus/Golgi; connects cell membrane constituents to the actin cytoskeleton
Ganglioside-Induced Differentiation-Associated Protein 1 (GDAP1)	D (brain)	57–201		Mitochondria/nucleus/cytoplasm; may play a role in neuronal development
Human Immunodeficiency Virus Type I Enhancer-Binding Protein 1 (HIVEP1)	D (brain)	1947–2377		Mitochondria/nucleus/cytoplasm; transcription factor
Heterogeneous Nuclear Ribonucleoprotein L (HNRPL)	D (brain)	371–480		EC/nucleus/cytoplasm; splicing factor
Kinesin Family Member 5A (KIF5A)	D (brain)	767–953		Cytoskeleton/cytoplasm/nucleus; microtubule motor protein
Microtubule–Actin Crosslinking Factor 1 (MACF1)	D (brain)	4485–4695		Cytoskeleton/Golgi/plasma membrane/cytoplasm; facilitates actin microtubule interactions
Mitochondrial Ribosomal Protein S9 (MRPS9)	D (brain)	100–396		Mitochondria/nucleus/cytoplasm; mitochondrial protein synthesis
PLAG1-Like Zinc Finger 1 (PLAGL1)	D (brain)	1–215		Golgi/nucleus; transcription factor that functions as a suppressor of cell growth
RAN-Binding Protein 3-Like (RanBP3L)	D (brain)	273–490		Nucleus/cytoplasm; intracellular transport. NT.
SH3 Domain-Binding Protein 5 (SH3BP5)	D (brain)	56–125		Mitochondria/nucleus/cytoplasm; inhibits the auto- and transphosphorylation activity of Bruton’s tyrosine kinase
Tousled-Like Kinase 1 (TLK1)	D (brain)	1–305		Nucleus/cytoplasm; serine/threonine kinase, which may be involved in the regulation of chromatin assembly
Zinc Finger Protein 419 (ZNF419)	D (brain)	73–320		Nucleus/cytoplasm; may be involved in transcriptional regulation
Zinc Finger Protein 483 (ZNF483)	D (brain)	520–681		Nucleus/cytoplasm; may be involved in transcriptional regulation
CREB/ATF BZIP Transcription Factor (CREBZF)	E (lung)	183–304		Nucleus/cytoplasm/mitochondria; transcription factor
X-ray Repair Cross-Complementing 6 (KU70; XRCC6; G22P1)	E (brain, testis)	477–609	Baade et al., 2018	Nucleus/cytoplasm/mitochondria; forms dimer with KU80, together act as the regulatory subunit involved in non-homologous end-joining required for DNA double-strand break repair/DNA helicase/suppression of apoptosis
Hes-Related Family BHLH Transcription Factor with YRPW Motif 2 (HEY2)	E (testis)	18–187		Nucleus/plasma membrane; transcriptional repressor implicated in cardiovascular development, neurogenesis, and somitogenesis/part of Notch signalling pathway.
Protein Phosphatase 2 Regulatory Subunit B’Epsilon (PPP2R5E)	E (lung, testis)	108–374	Baade et al., 2018	Cytoplasm/nucleus; regulatory subunit of the phosphatase 2A serine/threonine phosphatase implicated in the negative control of cell growth and division
Hes-Related Family BHLH Transcription Factor with YRPW Motif 1 (HEY1)	E (brain, lung)	32–150		Nucleus/plasma membrane; transcriptional repressor/part of Notch signalling pathway.
High Mobility Group 20A (HMG20A)	E (brain, lung)	31–198	Fatima et al., 2017	Nucleus; transcription factor
Zinc-Finger- and BTB Domain-Containing 38 (ZBTB38)	E (lung)	431–784		EC/nucleus; zinc-finger-containing transcriptional activator
CXXC Finger Protein 5 (HSPC195)	E (brain)	135–227		Nucleus/cytoplasm; retinoid-inducible protein involved in myelopoiesis/required for DNA damage-induced p53 activation/regulates the differentiation of myoblasts into myocytes/negatively regulates cutaneous wound healing
Paired Box 6 (PAX6)	E (brain)	106–346	Ploski et al., 2004	Nucleus/cytoskeleton/cytoplasm/EC; regulator of transcription
Tripartite Motif-Containing 26 (TRIM26)	E (brain)	123–284		Nucleus/cytoplasm; may have DNA-binding activity
Zinc Finger Protein 133 (ZNF133)	E (brain)	129–398		Nucleus/cytoplasm; may be involved in transcriptional regulation as a repressor

## Data Availability

Not applicable.

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
