# Peer review of "The Nuclear Transporter Importin 13 Can Regulate Stress-Induced Cell Death through the Clusterin/KU70 Axis"

_cells, 2023, doi:10.3390/cells12020279_

Round 1
Reviewer 1 Report
This manuscript describe roles of Importin 13 under stress condition through mediating transport of Clustrin/KU70. The overall study seems to be interesting and important, but I cannot judge the manuscript due to lack of Figures. For example, only (a) to (i)is presented in Figure 1, there is no (h) to (o) in Figure 1 as described in the legend and text. We don't see any results of Imp7 in Figure 5 as described. I cannot judge such immature manuscript.
Besides lack of Figures, I have comments that I hope authors can answer.
1) Heat stress and oxidative stress are different, but authors seems to think that they are same: for example, they describe transport of many Importins are equally affected by heat stress and oxidative stress due to defects of Ran. Does author consider Importin alpha/beta and Imp 7 is affected by stress because of Ran defect, but Imp 13 is not affected by Ran defect? Is this correct?
2)In Fig5 a, I don't see the DAPI staining. In case of + 43C, does Ran completely exit the nucleus or are they absent from middle of nucleus but stay on the nuclear periphery?
3) Please explain more clearly, what cause the difference between Imp7 and Imp13?
4)The authors clearly indicate that Imp 13 dependent nuclear transport of nCLU and KU70 are not affected under stress condition. This observation is interesting and important. Is this specific to substrates (nCLU and KU70) or is Imp13 transport in general is not affected by stress? Please clarify this point, with additional date if necessary.
Reviewer 2 Report
Gajewska et al. showed the nuclear transporter Importin 13 can regulate stress-induced 2 cell death through the Clusterin/KU70 axis. The biggest problem of this paper is lacking appropriate controls. The authors at least need to prove of other nuclear transporter karyopherins (Imp 1-7) do or do not reduce stress-induced import of nCLU and protecting from stress-induced cell death or not.
They also need to perform overexpression of various karyopherins and evaluate the effects of nCLU localization before drawing the conclusion.
Reviewer 3 Report
This manuscript by Gajewska et al., describes the discovery that IPO13, a bidirectional transportin than releases cargo in a Ran independent manner, continues to function under conditions of cellular stress that shut down the Ran cycle and Ran-dependent transportins. nCLU is imported during stress but KU70 is not exported, both normally cargo of IPO13. Loss of IPO13 protects cells from stress-induced apoptosis and DNA damage. I found this manuscript a bit difficult to interpret in places, but was able to largely follow the experimental design, results and interpretation of results. There are some places that I had concerns, including in data presentation noted below. This manuscript should be of considerable interest to those studying nuclear transport and cell stress.
Specific comments:
The use of overexpression of IPO13 in a cell that already expresses IPO13 is confusing. Is transport typically mediated by levels of tranportins, or is it regulated by signals/modifications that mediate transporting-cargo interactions? This seems like an unexpected mechanism of regulating transport.
Why not use FLIP to assess transport as this is a more logical approach to assess the movement of proteins between (as opposed to within) compartments. FRAP is best used for recovery of proteins to a bleached area, not loss from an unbleached area.
For result of fig 2 E-F it states compared to untreated DsRed2 co-expressing 544 cells, GFP-KU70 showed significant (p < 0.0161) 4-fold loss of nuclear fluorescence in cells 545 treated with H2O2 (Maximal Loss of Fluoresence = -0.75 and -0.2; Fig. 2g), consistent with the idea that oxidative stress reduces KU70 nuclear export,. How is a loss of nuclear fluorescence being 4-fold more than in untreated cells a sign that export is reduced? Maybe the control cells have a 4-fold loss compared to peroxide treated cells. That makes more sense and is probably what the data shows.
In fig 2E, the representative peroxide treated cell that is shown does not appear to clearly photobleach, making any assessment difficult.
Should the mobility of the cargos be assessed under each condition in the respective compartments? For example, KU70 may bind damaged DNA (or DNA repair machinery) more stably in the presence of DNA damage, thus the ability of the transportin to export the protein is reduced.
Figure 1 has a mixture of the more appropriate inclusion of all data in 1b, but a representative sampling of data in 1d. All data should be evaluated and presented together. Not selectively and variably.
Figures 2, 6 and S4 doesn’t indicate if there were experimental replicates. There is N>20, but not mention of replicates to reach that N.
In Figures 3, 5, S1, S2, S3 there is mention of single replicate results being shown, but that replicates were performed. Why not show the aggregate data from the three replicates? And/or quantify the average changes between relevant conditions for each replicate.
Does IPO13 continue to shuttle with depletion of GTP from the cell, e.g. by treating with mycophenolic acid (MPA)? MPA will shut down the Ran cycle, so this would be a direct way to address if IPO13 continues to function even if the Ran cycle is effectively ablated. How about ATP depletion, would that inhibit IPO13 mediated transport?
Round 2
Reviewer 2 Report
I have no further questions.